# Genomic profiling of subcutaneous patient-derived xenografts reveals immune constraints on tumor evolution in childhood solid cancer

Funan He [1,2,12], Abhik M. Bandyopadhyay[1,12], Laura J. Klesse [3,4,5], Anna Rogojina[1], Sang H. Chun[6], Erin Butler[3,4,5], Taylor Hartshorne [3], Trevor Holland[1], Dawn Garcia[1], Korri Weldon[1], Luz-Nereida Perez Prado[1], Anne-Marie Langevin[7,8], Allison C. Grimes[1,7], Aaron Sugalski[7], Shafqat Shah[7], Chatchawin Assanasen[7,8], Zhao Lai[1,8,9], Yi Zou[1], Dias Kurmashev[1], Lin Xu [3,4,10], Yang Xie[4,10,11], Yidong Chen[1,2,8], Xiaojing Wang [1,2,8], Gail E. Tomlinson[1,7,8], Stephen X. Skapek[3,4,5], Peter J. Houghton[1,8,9], Raushan T. Kurmasheva [1,8,9] ✉ & Siyuan Zheng [1,2,8] ✉

Subcutaneous patient-derived xenografts (PDXs) are an important tool for childhood cancer research. Here, we describe a resource of 68 early passage PDXs established from 65 pediatric solid tumor patients. Through genomic profiling of paired PDXs and patient tumors (PTs), we observe low mutational similarity in about 30% of the PT/PDX pairs. Clonal analysis in these pairs show an aggressive PT minor subclone seeds the major clone in the PDX. We show evidence that this subclone is more immunogenic and is likely suppressed by immune responses in the PT. These results suggest interplay between intratumoral heterogeneity and antitumor immunity may underlie the genetic disparity between PTs and PDXs. We further show that PDXs generally recapitulate PTs in copy number and transcriptomic profiles. Finally, we report a gene fusion LRPAP1-PDGFRA. In summary, we report a childhood cancer PDX resource and our study highlights the role of immune constraints on tumor evolution.

Childhood cancers represent about 1% of newly diagnosed cancer cases in the US. Though rare, cancer is the leading cause of disease-related death in children[1]. More than 60% of childhood cancer cases are solid tumors. The average five-year survival rate for children with solid cancers exceeds 80%, but survival for patients with metastatic or refractory tumors is still poor. Further, multimodality treatments cause long-term health problems and increase the risk of secondary cancer[2,3]. Molecularly targeted therapies and immunotherapy can improve overall patient outcomes, but their development requires faithful preclinical models and a better understanding of antitumor immunity.

Patient-derived xenografts (PDXs) are an important model in cancer research. They are crucial for preclinical and mechanistic studies of rare cancers such as pediatric solid tumor because they can preserve tumor tissue in vivo[4–7]. PDXs are established by engrafting tumor tissue either subcutaneously or orthotopically into immuno-compromised mice. Compared with orthotopic PDXs, subcutaneous PDXs are easier to establish and monitor tumor size. Preclinical testing studies with subcutaneous PDXs showed that they can robustly inform drug activity in patients[8,9].

A fundamental question about PDXs is how well they recapitulate the patient tumors (PTs). In adult cancers, PDXs were found to

recapitulate PTs in histology, genetics, and pharmacokinetics[10–13]. However, genomic profiling of large PDX cohorts found evidence of clonal evolution during engraftment and passaging, leading to debates over model fidelity[14–16]. In childhood solid cancers, similar genomic profiling efforts were undertaken, but often without matched PTs or germline samples[17]. Other studies were focused on single cancer types[18–20], or orthotopic models[21,22], or with a limited sample size[23–25]. Moreover, many rare childhood cancers such as hepatoblastoma were often not included. Importantly, both adult and childhood cancer studies have found PDXs that showed poor mutational similarities with PTs[18,21,26], but the underlying mechanism leading to the disparity remains obscure.

Here, we report genomic profiling of 68 solid childhood cancer subcutaneous PDXs. These models were established from 65 pediatric solid tumors across 16 cancer types.

## Results

### Overview of patient samples, PDXs, and genomic data

We generated 90 subcutaneous PDXs from 194 fresh solid tumor samples using a previously published protocol[27] ("Methods"). All patients were younger than 18 years old at the time of tumor collection, with both biological sexes represented (Male:Female, 1.2:1). Of the patients with treatment information available, 38% received prior treatment, primarily chemotherapy.

We observed high engraftment rates in clear cell sarcoma (100%) and Wilms tumor (85%), and lower rates in neuroblastoma (26%) and brain tumors (23%) (Fig. 1a). The engraftment rates for neuroblastoma and Wilms tumor were similar to that of previously published orthotopic models[21], but the rate for osteosarcoma in our cohort was higher (67% vs. 48%). The average time from tumor implantation (P0) to PDX harvest (P1) also varied from 30 weeks for neuroblastoma to 13 weeks for hepatoblastoma (Fig. 1b).

We performed low pass whole genome sequencing (WGS), whole exome sequencing (WES), and mRNA sequencing (RNAseq) on 68 PDXs (Fig. 1c). Among the 68 PDXs, 27 (40%) had the matched patient tumor (PT) and 40 (59%) had normal germline DNA. Isogeneity of the matched samples was confirmed using DNA and RNA sequencing data (Supplementary Fig. 1). The PDX cohort comprised 14 Wilms tumors, 13 hepatoblastomas, 12 osteosarcomas, 10 germ cell tumors, and 19 others. These PDXs were derived from tumor tissues of 65

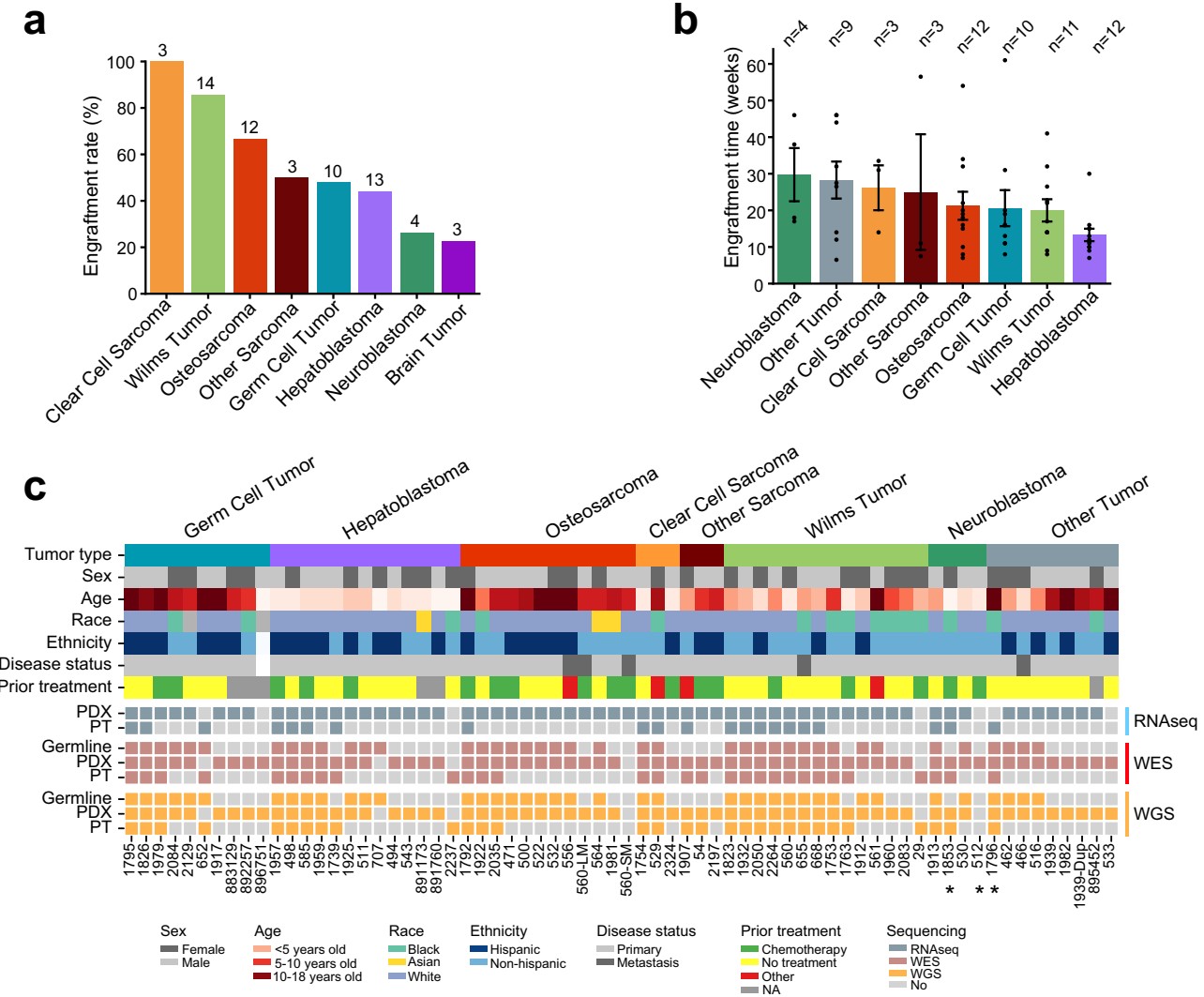

**Fig. 1 | Overview of PDXs and sequencing data. a** Engraftment rate across cancer types. The number above each bar indicates the number of PDXs analyzed in this study. **b** Average engraftment time. The error bar indicates standard error of the mean. **c** Overview of clinical and molecular data. The top panel shows clinical data including cancer type, stage, sex, age, race, ethnicity, and treatment. The bottom panel summarizes the sequencing data. Samples with an asterisk were removed from data analyses because of high mouse tissue contamination. RNAseq, RNA sequencing; WES, whole exome sequencing; WGS, low pass whole genome sequencing. Source data are provided as a Source data file.

patients, all younger than 18 years (median 6.5; mean 7.7). Male to female ratio (1.3:1) was similar to the overall patient cohort ($n = 90$, 1.2:1). Thirty patients were of Hispanic ancestry, and this was confirmed using ancestry informative markers[28] (Supplementary Fig. 2). Treatment information was collected for 62 patients, 23 of whom received prior treatment (37%). Sixty-two models were derived from primary tumors and five from metastatic tumors. Clinical data of the samples were summarized in Supplementary Data 1. To help disseminate this resource, we have built an intuitive online portal (pediatric solid tumor PDX portal, https://pstPDX.streamlit.app). Requests for PDX materials can also be made on the site.

We observed higher tumor purity in PDXs than in PTs ($p = 2.1 \times 10^{-3}$, two-sided $t$ test; Supplementary Fig. 3a, b). Immune and stromal cell signature scores were lower in PDXs (Supplementary Fig. 3c–e). Interestingly, the signature scores of stromal cells in PDXs were positively correlated with those in PTs, suggesting early passage PDXs still retained some stromal cells from the PT (rho = 0.66, $p = 0.003$; Spearman correlation). We did not observe significant correlation for immune scores (Supplementary Fig. 3f).

## Mutational similarity between PT and PDX

We used multiple tools to detect somatic mutations and indels (insertions and deletions) (Methods). In total, we identified 1786 mutations and 161 indels from WES data. Ninety-two percent of the point mutations were validated in RNAseq or low pass WGS. The unvalidated mutations had lower variant allele fractions (VAFs, mean 0.13 vs. 0.34 for validated mutations). Deep sequencing of a PT/PDX pair yielded a 100% validation rate on 60 mutations that were captured in the assay (Supplementary Data 2). We did not observe significant differences in mutation rate between PTs and PDXs except in Wilms tumor, or tumors with and without a germline control except in germ cell tumor (Supplementary Fig. 4a, b). Across the cancer types, Wilms tumor showed the lowest mutation rate (median: 0.18 mutations/Mb), and osteosarcoma showed the highest (median: 0.56 mutations/Mb) (Fig. 2a). These mutation rates agree with results from recent pan-pediatric cancer analyses[29,30].

Few cancer genes showed recurrent mutations across the cohort, consistent with the overall low mutation rate of childhood cancers (Supplementary Fig. 4c). The exception was *CTNNB1*, which was mutated in 7 of 11 (64%) hepatoblastomas with exome sequencing data. Mutation rates of known driver genes from our dataset were generally consistent with the literature (Supplementary Data 3). Pan-cancer analysis with MutsigCV[31] identified only *CTNNB1* and *TP53* as significant mutated genes across the PDXs (FDR < 0.1) (Supplementary Data 4).

We observed significantly higher mutation rates in prior treated PTs or their derived PDXs than in treatment-naïve samples ($p = 2.3 \times 10^{-4}$, Wilcoxon rank sum test; Fig. 2a). To corroborate the association between higher mutation rates and chemotherapy, we deconvoluted mutations into mutational signatures. Such deconvolution can identify mechanisms that cause mutations in the cancer genome[32]. We in total analyzed 14 samples with at least 20 mutations (Fig. 2b). Among the 10 samples that were derived from patients who had received chemotherapy, we found evidence of chemotherapy related mutational signatures in 8 (2 PTs and 6 PDXs), seven associated with the platinum drug related signatures SBS31 and SBS35, and one with SBS86, a signature currently associated with unknown chemotherapy treatment. The samples demonstrating signature SBS31 or SBS35 were derived from six patients. Except for an osteosarcoma patient (560-LM) who received unspecified chemo-treatment, all the other five patients received cisplatin, a platinum-based drug. This data supports the mutational signature analysis. The sample not exhibiting chemotherapy signatures (1981_PDX) showed SBS15, a signature associated with microsatellite instability (MSI). Consistently, the sample showed a high MSI score (42.4% vs. average of all others, 1.6%).

For the two PTs that exhibited high mutation rates and chemotherapy signatures (1792_PT, 1957_PT), their corresponding PDXs also exhibited the same signatures (Fig. 2b). Another PT sample, 585_PT, was dropped from mutational signature analysis due to its low mutation count ($n = 6$); however, the matched PDX exhibited SBS31 and SBS35. The consistency in demonstrating chemotherapy signatures was not necessarily driven by shared mutations between PTs and PDXs. For 585 and 1957, PTs and PDXs had little overlap in somatic mutations (Fig. 2c). Using PDX-specific mutations yielded the same chemotherapy signatures for the two samples (Supplementary Fig. 4d). Thus, these data suggest the related mutations in these PDXs were inherited from the seeding PTs.

Next, we examined mutational similarities between PTs and PDXs using 25 PT/PDX pairs (Fig. 2c). We defined mutational similarity as the fraction of shared mutations over all mutations found in each pair. Overall, 78% of mutations were shared between PTs and PDXs. The median mutational similarity was 0.52, higher than those observed in recently published pediatric cancer PDX cohorts[18,21] but lower than that in adult tumors[26]. Limiting the comparison to cancer genes (Supplementary Data 5) increased the median mutation similarity to 0.95 for the 20 pairs where at least one cancer gene mutation was observed (Supplementary Fig. 4e, f). Oncogenic or likely oncogenic mutations demonstrated a high level of overlap (28/30, 93%) between PTs and PDXs. Five pairs showed low mutational similarity (<0.2), including two (1959, 1979) with no shared mutations. To test if the low mutational similarity was due to sequencing coverage, we performed capture enrichment and deep sequencing on a pair of PT and PDX samples (585_PT and 585_PDX). By WES, six mutations were found in 585_PT, and 58 mutations were found in 585_PDX, 56 of which were not found in 585_PT. Deep sequencing captured 54 mutations found in 585_PDX, all validated. Similarly, all six mutations found in 585_PT were validated by deep sequencing (Supplementary Data 2). None of the PT or PDX specific mutations were found in the matched sample in the deep sequencing data, suggesting limited impact by sequencing depth on the observed PT/PDX mutational similarity in this case.

To understand how intratumoral heterogeneity can impact PT/PDX mutational similarity, we obtained seven additional PDX samples that matched four PTs. Six of the seven PDXs were established from a distinct patient tumor block, and the remainder was a second block of the originally sequenced PDX. Comparison of these additional PDXs with matched PTs demonstrated generally consistent mutation similarities in these samples (Supplementary Fig. 4g).

## Distinct evolutionary patterns during engraftment

To explore the clonal dynamics in tumor engraftment, we inferred mutation clonality using a consensus approach for the 25 PT-PDX pairs ("Methods"). Overall, 82% of mutations in PTs and 84% of mutations in PDXs were clonal, but this percentage was highly case specific (Supplementary Fig. 5). While 88% of PT clonal mutations were observed in the PDX, only 22% of PT subclonal mutations were observed in the PDX. This result was consistent with the expectation that clonal mutations more likely pass on than subclonal mutations. To further validate mutation clonality, we examined presence of PT clonal and subclonal mutations in the additional PDXs. For the four PTs with multiple PDXs, all PT clonal mutations ($n = 30$) that were observed in the original PDX were also observed in the additional PDXs. In contrast, only two of the 17 PT subclonal mutations were observed in the additional PDXs. Notably, 33% of PDX clonal mutations were not found in the PT, suggesting clonal expansion during engraftment (Supplementary Fig. 5a).

We next classified paired samples into distinct evolutionary patterns based on changes in mutation clonality from PTs to PDXs. For this analysis, we excluded the two PT/PDX pairs (1959 and 1979) that showed no mutational overlaps. We observed three patterns. In the first pattern, PDXs retain clonal mutations from the PT and

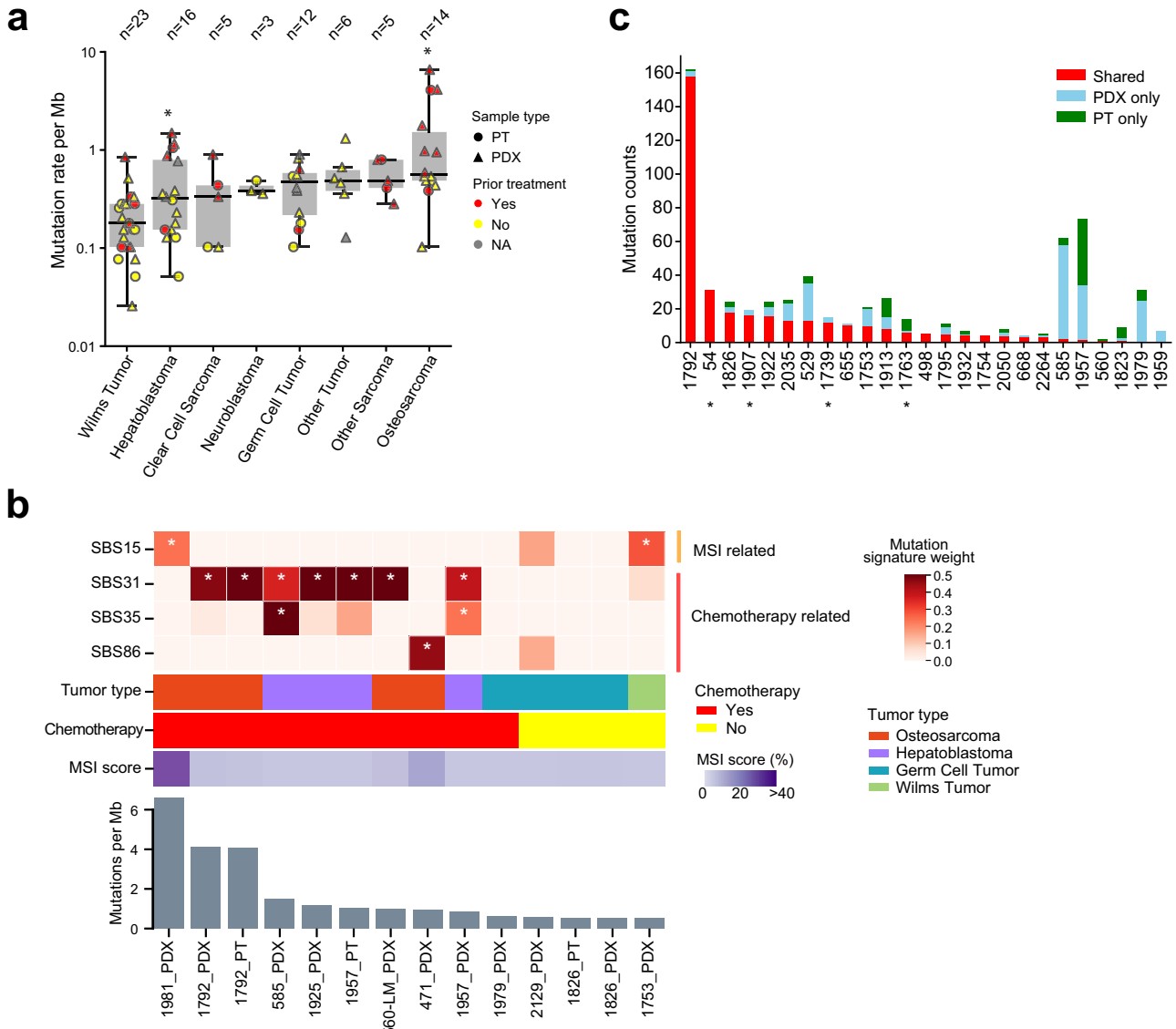

**Fig. 2 | Mutation rates and mutational signatures. a** Distribution of somatic mutation rate. Each dot represents a sample (circle, PT; triangle, PDX). The boxes and middle lines within depict the interquartile range (IQR) and median. The top and bottom whiskers represent values within 1.5×IQR of the upper and lower quantiles, respectively. In hepatoblastoma and osteosarcoma, samples with prior treatment show higher mutation rates than samples without (asterisk, $p < 0.05$, two-sided Wilcox rank sum test). Other tumor types either show no statistical significance ($p > 0.05$, Wilms tumor and germ cell tumor) or were not compared due to limited sample sizes. **b** Distribution of MSI and chemotherapy related mutational signatures across samples. The heatmap on top reflects the weight of selected mutation signatures. Clinical data and MSI score are shown below the heatmap. Bar plot at the bottom illustrates mutational load of each sample. Samples are ordered by mutation load. Asterisks indicate weight greater than 0.25. **c** Mutational similarity between PTs and PDXs. Patients are ordered by the total number of shared mutations. Samples labeled with asterisk do not have the matched germline. Source data are provided as a Source data file.

exhibit a similar clonal composition. We call this pattern 'clone retention' (Fig. 3a and Supplementary Fig. 5b). This pattern constituted 70% (16/23) of the pairs classified. The second pattern was characterized by expansion of PT subclones in the PDX (Fig. 3b and Supplementary Fig. 5c). This pattern, termed "clone sweeping," was observed in four pairs (17%). The last pattern was characterized by loss of PT clonal mutations and retention of early mutations in the PDX (Fig. 3c and Supplementary Fig. 5d). This pattern, termed 'branch seeding', was observed in three pairs (13%). The loss of PT clonal mutations was not due to copy number deletion in the paired PDX. One example of this pattern was a hepatoblastoma sample (1957); only two of the 41 PT mutations were found in the PDX, one of which was in *CTNNB1*, an early driver of the cancer type[33] (Fig. 3c).

The evolutionary patterns appeared to be reproducible across multiple PDXs. In two samples (1913,1932) that were classified as clone

sweeping, the evidence for these classifications were that in both cases, a PT subclonal mutation became a clonal mutation in the PDX (*LRP2* for 1913, and *BMP4* for 1932). Interestingly, the same *LRP2* mutation was also identified in the two additional 1913 PDXs where the mutation appears to be clonal (VAF 0.44 and 0.45, vs. 0.09 in PT). Similarly for 1932, the *BMP4* mutation was observed in the two additional PDXs, also with much higher VAFs in the PDXs (0.37 and 0.38 vs. 0.12 in PT).

Both patterns of clone sweeping and branch seeding indicate that a subclone in the PT seeds the PDX, likely by outcompeting other clones. For simplicity, we lump them together as one group (group 2), to compare with samples showing the clone retention pattern (group 1). Unlike continued expansion of the major PT clone in the PDX (group 1), clonal selection observed in group 2 would take longer to establish a major clone. Consistent with this idea, the median time for group 2

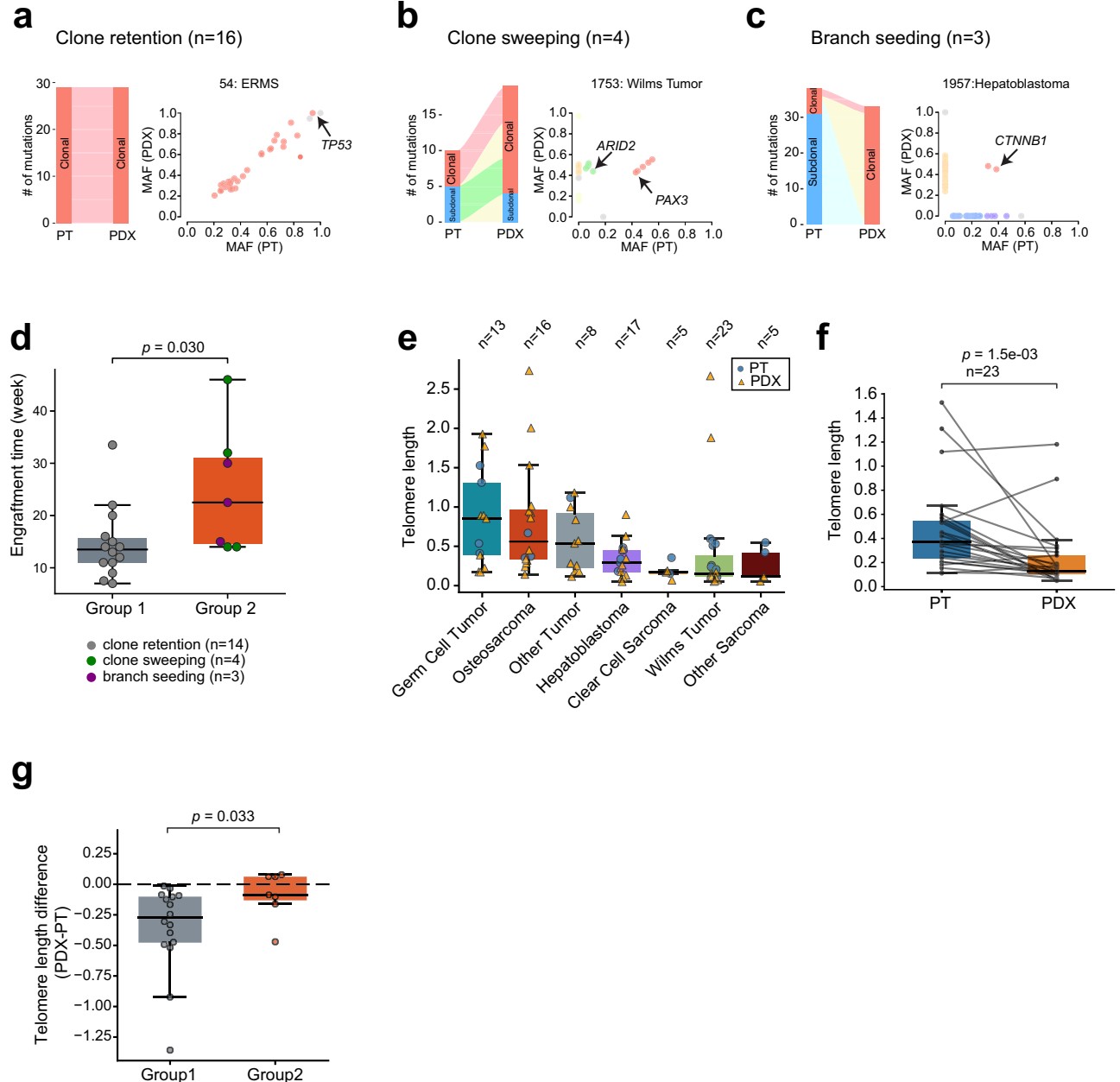

**Fig. 3 | Evolutionary patterns from PTs to PDXs. a** A sample that shows clone retention. Left, sankey plot showing mutation clonality flow from the PT to PDX. The width of the joining lines reflects the number of mutations. Right, scatter plot of tumor mutant allele fraction (MAF) between the PT (*x* axis) and PDX (*y* axis). Cancer-related genes are highlighted with arrows. Similarly, **b** clone sweeping, **c** branch seeding. **d** Longer engraftment time for group 2 tumors. Each dot represents one PDX. *Y* axis represents the time in weeks from initial implantation to harvest. *P*-value was calculated with two-sided Wilcoxon rank sum test. The box represents the interquartile range (IQR), and middle line represents median. Whiskers represent values within 1.5×IQR of the upper and lower quantiles respectively. **e** Telomere length across cancer types. Each dot represents one sample. Boxplot is interpreted similarly as above. **f** Comparison of telomere length between paired PTs and PDXs. The box is interpreted similarly as above. *P*-value was calculated with paired two-sided Wilcoxon rank sum test. **g** Longer relative telomere lengths in group 2 PDXs. *Y* axis represents the difference in telomere length for paired PT and PDX. The dashed line indicates no telomere length difference between PT and PDX. *P*-value was calculated with two-sided Wilcoxon rank sum test. Boxplot is interpreted similarly as above. Source data are provided as a Source data file.

models to reach the harvest tumor volume after implantation was 22 weeks, compared to 13 weeks for group 1 models (*p* = 0.03, Wilcoxon rank sum test; Fig. 3d).

The longer engraftment time could explain the increased number of PDX specific mutations in group 2 (Fig. 4a). To test this possibility, we correlated the two and found no significant correlation (*p* = 0.18, Spearman correlation test; Supplementary Fig. 5e). This lack of correlation remained after controlling for the PT mutation rate for each PDX (*p* = 0.13; Supplementary Fig. 5f).

To provide further evidence for the distinct evolutionary paths, we analyzed tumor telomere lengths. Telomeres progressively shorten along cell divisions[34]; thus, continued growth of the same clones from PT to PDX such as in group 1 would likely result in shorter telomeres in the PDX. We estimated average tumor telomere lengths using both WGS and WES data (Methods). The two data types yielded consistent telomere length estimates in PDXs, PTs, and germline samples (Supplementary Fig. 6a and Supplementary Data 6). Across the cancer types, germ cell tumor showed the longest telomeres (Fig. 3e and

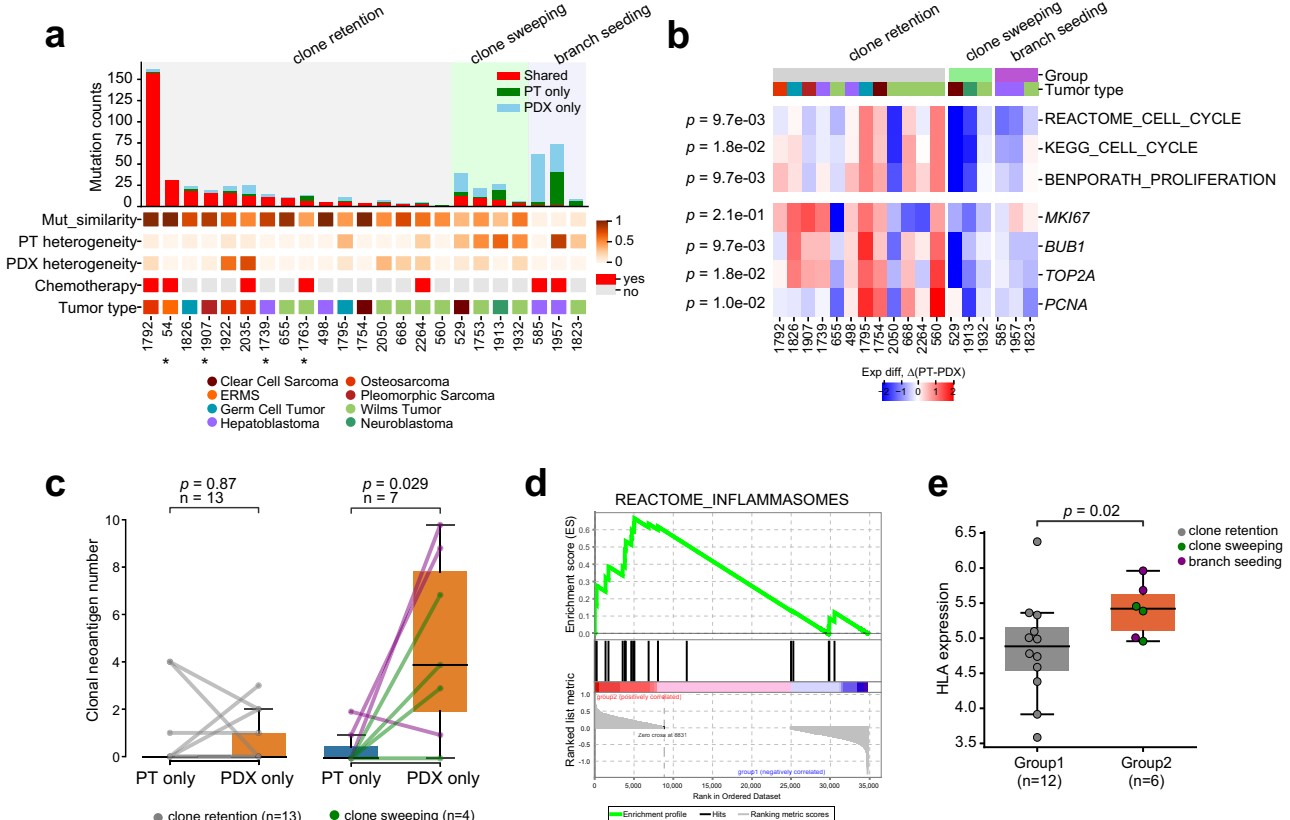

**Fig. 4 | Evolutionary pattern is associated with mutational similarity and antitumor immunity in PT. a** The evolutionary patterns are correlated with genetic heterogeneity of the PT. Patients are ordered by the number of shared mutations between PT and PDX within each pattern. The top bar plot shows mutation overlap for each PT/PDX pair. The bottom panel shows mutational similarity, PT genetic heterogeneity, PDX genetic heterogeneity, chemotherapy, and cancer type. Samples labeled with asterisk do not have the matched germline. **b** Difference in expression of proliferation markers and cell cycle signatures. Red, higher in PT; Blue, higher in PDX. The signatures were scored with ssGSEA. The *p*-values on the left were calculated between group 1 and group 2 using two-sided Wilcoxon rank sum test. **c** Changes in clonal neoantigen count between PTs and

PDXs. The left panel shows the changes for group 1 samples, and the right panel shows those for group 2 samples. Each dot represents one sample. *P*-values were calculated with two-sided paired *t* test. The box represents the interquartile range (IQR), and middle line represents median. Whiskers represent values within 1.5×IQR of the upper and lower quantiles respectively. **d** Pathway analysis identifies inflammasome pathway higher in group 2 PTs compared with group 1 PTs (*p* = 0.03). *P*-value is reported as is from Gene Set Enrichment Analysis. **e** HLA genes, which encode MHC complexes on APC surfaces, are highly expressed in group 2 PTs (*p* = 0.02, one-sided Wilcoxon rank sum test). HLA gene expression is summarized using ssGSEA. The boxplots are interpreted similarly as in panel c. Source data are provided as a Source data file.

Supplementary Fig. 6b), likely due to its origin from telomerase-competent germ cells. Among non-germ cell cancers, osteosarcoma showed the highest telomere length, consistent with a recent report[35].

PDXs showed overall shorter telomeres than matched PTs ($p = 1.5 \times 10^{-3}$, paired t test; Fig. 3f and Supplementary Fig. 6c). The pattern of telomere shortening was more pronounced in group 1 than in group 2 tumors (*p* = 0.033, Wilcoxon rank sum test; Fig. 3g and Supplementary Fig. 6d). These data provide additional evidence that group 2 tumors underwent a distinct evolutionary path from group 1 tumors.

**Clonal selection during engraftment associates with genetic heterogeneity and antitumor immunity in the PT**
To provide insights into the three evolutionary patterns, we correlated them with PT/PDX mutational similarity, prior treatment, and tumor genetic heterogeneity defined as the fraction of subclonal mutations in a sample. 'Clone retention' tumors showed an average mutational similarity of 0.73, compared to 0.42 for 'clone sweeping' and 0.06 for 'branch seeding' (group 1 vs. group 2, *p* = 0.0012, Wilcoxon rank sum test). The decreasing mutational similarity was associated with PT (*p* = 0.034, Spearman correlation; Supplementary Fig. 7a) but not PDX genetic heterogeneity (*p* = 0.38, Fig. 4a). Thus, PTs with more complex

clonal structures tend to generate genetically more distinct subcutaneous PDXs. We did not find a significant association between chemotherapy and the evolutionary patterns (*p* = 1, chi-square test).

We next asked what drove the clonal selection in group 2 tumors. For these pairs, we denote the major clone in the PT as $C_{pt}$ and the major clone in the PDX as $C_{pdx}$. Based on their evolutionary pattern, clone $C_{pdx}$ was a minor clone in the PT. We hypothesized that $C_{pdx}$ had a growth advantage so that it could overtake $C_{pt}$ during engraftment. To test this hypothesis, we compared proliferation markers and cell cycle signatures between PTs and PDXs, assuming expression of bulk tumor reflected the major clone's. Consistent with the hypothesis, expression of proliferation markers and cell cycle signatures was higher in PDXs than in the matched PTs for group 2 tumors, suggesting the PDX major clone was indeed more proliferative in these pairs. In contrast, the opposite pattern was observed in group 1 tumors (Fig. 4b).

If $C_{pdx}$ was more proliferative, why was it not the major clone in the PT? We reasoned that its expansion could be constrained in the PT, but such constraint was weakened or even nullified in the PDX. Because PDX-host mice have no functional immune system, immune surveillance may contribute to this constraint by preferentially targeting $C_{pdx}$.

To test this hypothesis, we first compared mutational load. For both groups, no significant difference in mutational load was observed between PTs and PDXs (Group 1, $p = 0.2$; Group 2, $p = 0.3$, paired $t$ test; Supplementary Fig. 7b). We next compared neoantigens (Supplementary Data 7; "Methods"). In group 1, no difference was found in clonal neoantigen load between PTs and PDXs ($p = 0.87$, paired $t$ test). However, in group 2, PDXs showed significantly more clonal neoantigens than their matched PTs ($p = 0.03$, paired $t$ test; Fig. 4c). This pattern remained after controlling for the total number of clonal mutations ($p = 0.03$, paired $t$ test; Supplementary Fig. 7c). Thus, despite being more proliferative, $C_{pdx}$ also expressed more neoantigens.

There was evidence that subclonal neoantigens are more immunogenic than clonal ones[36]. Because $C_{pdx}$ was a subclone in the PT, we sought to find signs of antitumor immunity in group 2 PTs. We compared their expression with those from group 1 PTs. Pathway analysis with GSEA identified inflammasomes as the second highest pathway ranked by normalized enrichment score in group 2 PTs ($p = 0.03$; Fig. 4d and Supplementary Data 8). Inflammasomes are the receptors and sensors of the innate immune system[37]. They are assembled in professional antigen-presenting cells (APCs), which are constituents of the innate immunity and bridge the innate and adaptive immune systems[38]. High inflammasome activity suggests possible activation of the innate immunity. Consistently, gene signatures related to natural killer cells, a major effector cell of the innate immune system, were higher in group 2 PTs (Supplementary Fig. 7d). Expression of *HLA* genes, which encode major histocompatibility complexes (MHCs) on the surface of APCs, was significantly higher in group 2 PTs than group 1 PTs ($p = 0.02$, Wilcoxon rank sum test; Fig. 4e). We next examined tumor microenvironment using multiple deconvolution tools ("Methods"). Interestingly, the abundance of cancer associated fibroblast, a known immunosuppressive cell population[39], was consistently reported lower in group 2 PTs (Supplementary Fig. 7e).

Taken together, these results suggest in group 2 PT/PDX pairs, a more proliferative but also more immunogenic PT subclone was selected to seed the PDX. In the context of immune deficiency in the host mice and activated immune responses in the PT, these data implicate a role of immune environment changes in fostering this selection during engraftment.

## PDXs retain somatic copy number alterations

Whether somatic copy number alterations (SCNAs) undergo PDX specific evolution has been recently debated in adult cancer[14–16]. To examine SCNA conservation in our PDXs, we inferred copy number profiles using low pass WGS data (Methods). This data provides better resolution than exome sequencing-based estimates. Using these data, we identified 15 amplification and 19 deletion peaks (Supplementary Fig. 8a). Genes located in the amplification peaks included *MYC*, cell cycle genes *CCND3* and *CCNE1*, chromatin regulators *SETDB1* and *EZH2*, and DNA repair gene *XRCC2*. Genes located in deletion peaks included *TP53*, *PTEN*, DNA repair genes *RAD51*, *FANCA*, *ATM*, *CHEK1*, *POLD1*, apoptosis regulators *BAX* and *BCL2*, hypoxia regulator *HIF1A*, and interestingly *PD-L1*.

On the cancer type level, SCNA profiles were similar between PTs and PDXs, and were consistent with the literature (Supplementary Fig. 8b). For instance, we observed frequent gain of chr1q (57%) and loss of chr11 (30%) in Wilms tumor at a rate similar to previous reports[40,41]. Few SCNAs were observed in hepatoblastoma except arm-level gains of 1q (46%), 2q (41%), 20 (41%), and 8q (24%), as previously reported[42]. We observed frequent gain of chromosome 12p (85%), 21 (62%), 7p (54%), and loss of chromosome 4 (46%) and 5 (38%) in germ cell tumor. These rates were also consistent with previous studies[43,44].

To quantify SCNA conservation, we first compared tumor ploidy, a measure of genome wide SCNA. We found high tumor ploidy, likely driven by whole genome doubling (WGD), in 77% of germ cell tumors

and 67% of osteosarcomas (Fig. 5a). Tumor ploidy was highly similar between PDXs and PTs, including the group 2 tumors (rho = 0.98, $p = 3.2 \times 10^{-15}$, Spearman correlation; Fig. 5b), suggesting conservation of karyotype in the PDX. However, we did observe two exceptions (8%, patients 1959 and 1979) where drastic change in ploidy was found in the PDX. Consistent with this result, the PT and PDX of the two pairs did not show any overlap in mutations.

Next, we compared global chromosomal instability using a genomic instability (GI) score ("Methods"; Supplementary Data 9). Osteosarcoma, a cancer characterized by high chromosomal instability[45], showed the highest scores (Supplementary Fig. 9a). Tumors with relatively quiet genomes like neuroblastoma, clear cell sarcoma, and hepatoblastoma showed the lowest scores. GI scores were positively correlated between PTs and PDXs (rho = 0.75, $p = 1.7 \times 10^{-5}$, Spearman correlation; Supplementary Fig. 9b).

Finally, we correlated copy number profiles for each PT/PDX pair (Methods). After excluding samples with few SCNAs (total GI score <0.1), we found strong pairwise correlations between PTs and PDXs (Fig. 5c). Limiting this analysis to cancer genes yielded a similar result (Supplementary Fig. 9c). These strong correlations remained between PTs and multiple PDXs that were derived from the same patient tumor (Supplementary Fig. 9d). The correlations were similar between group 1 and group 2 tumors ($p = 0.6$, $t$ test; Supplementary Fig. 9e), suggesting SCNAs were primarily clonal. We then asked if focal events were retained in the PDX. In total we identified 292 focal events in nine samples, seven of which were sarcomas. The aggregated length of these events ranged from 2 Mb to 462 Mb (Supplementary Fig. 9f). Overall, the overlap of focal events was better than that of mutations, with 86% of them shared between PTs and PDXs (Fig. 5d). Unlike mutations, conservation of PT focal events in the PDX was observed in each pair analyzed. Only one PDX (2035) showed notably more private focal events. Taken together, these data show strong conservation of SCNAs in early passage PDXs.

## Transcriptomic analysis shows tissue effect and identifies fusions

We next examined how the PDXs recapitulated PTs in gene expression. Unsupervised clustering grouped samples into tissues of origin except clear cell sarcoma (Fig. 6a). Close analysis showed that clear cell sarcomas were divided into two groups, one consisting of samples collected from the kidney (1754 and 2324), and the other consisting of samples collected from the bone (529). This tissue-of-origin dominated pattern was previously observed in adult cancers[46].

We observed highly correlated expression profiles of the matched PTs and PDXs (rho range 0.92–1, Spearman correlation; Fig. 6b). To put these correlations in context, we compared PDXs that were derived from two metastatic lesions of the same patient (560 lung and skin metastases), and PDXs derived from different blocks of the same tumor (1939). The correlation between the two PDXs of patient 560 was 0.94, and the correlation between the two PDXs of patient 1939 was 0.97. The high correlation was similar across the three evolutionary patterns ($p = 0.87$, $t$ test; Supplementary Fig. 10a). Hepatoblastoma and Wilms tumor samples showed significant intra-lineage correlations, corroborating results from unsupervised clustering. These results show that gene expression is highly conserved in PDXs and is dictated by both cancer genetics and tissue of origin.

To identify molecular alterations, we called gene fusions using RNAseq data. We identified 161 high-confidence gene fusions (Supplementary Data 10; "Methods"), including disease-defining fusions such as reciprocal EWSR1-ATF1 in a clear cell sarcoma (patient 529) and BCOR-CCNB3 in a Ewing-like sarcoma (patient 2197). Most of the fusion events ($n = 125$, 78%) were found in osteosarcoma and clear cell sarcoma, and their distribution across cancer types was generally consistent with that of chromosomal instability (Supplementary Fig. 10b, c).

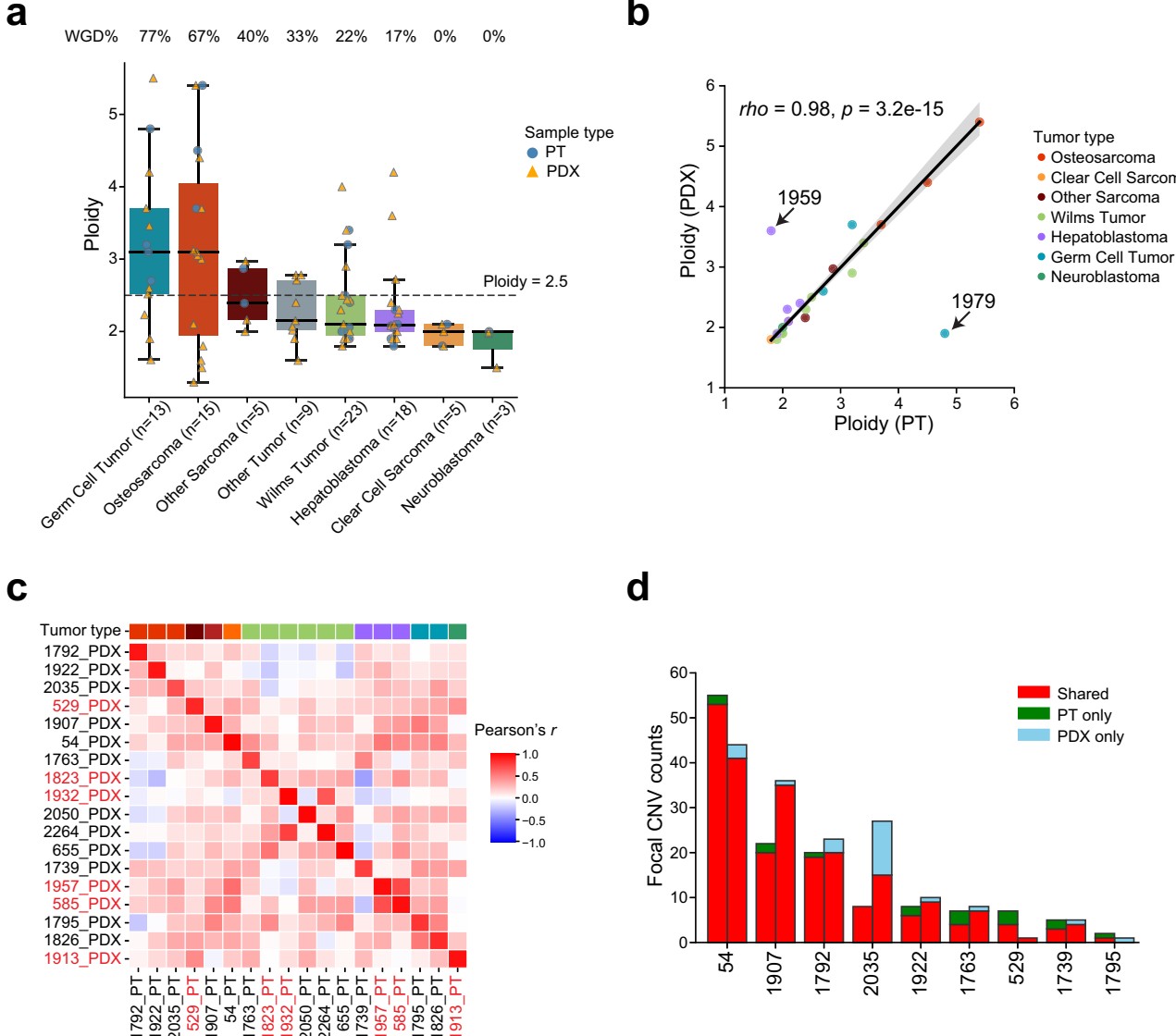

**Fig. 5 | Conservation of somatic copy number alterations (SCNAs) in PDXs.** **a** Distribution of tumor ploidy. Tumors with a ploidy higher than 2.5 (dashed line) are considered whole genome doubling (WGD). Each dot represents a sample (circle, PT; triangle, PDX). The boxes and middle lines within represent the interquartile range (IQR) and median. The top and bottom whiskers represent values within 1.5×IQR of the upper and lower quantiles respectively. **b** Ploidy correlation between PT and PDX. Each dot represents a pair. The correlation coefficient and *p*-value were calculated with Spearman's rank correlation test. Black solid line is the linear regression line, and the error band corresponds to 95% confidence interval. The two outliers (1959 and 1979) were excluded from the regression and correlation analyses. **c** Pairwise correlation in copy number profiles between PTs and PDXs based on 1 Mb windows. The top color bar indicating tumor type is interpreted the same as in (**b**). Samples labeled in red are from group 2. **d** Overlaps of focal SCNAs. For each patient, the left bar presents the PT and the right bar represents the PDX. Counts of shared events are different in PT and PDX because of variations in segmentation boundaries. Only the nine PT/PDX pairs where focal SCNAs were found are shown. Source data are provided as a Source Data file.

Paired PTs and PDXs showed significant overlaps in fusions. Of the 18 paired samples with RNAseq data, we identified at least one fusion in six pairs, and 97% (29/30) of the fusions detected in the PT were also found in the PDX (Supplementary Fig. 10d).

Next, we mapped the fusions to kinases and clinically actionable genes (Supplementary Data 10). Of the fusions identified in PDXs, 14 involved a kinase gene and 12 involved a clinically actionable gene, including TAOK1-NTRK3 (Fig. 6c). Inhibition of *NTRK* fusions showed promising clinical benefits in patients[47,48]. Importantly, we observed a fusion, LRPAP1-PDGFRA, in a glioblastoma and a germ cell tumor. The fusion preserved the protein kinase domain of PDGFRA and was associated with high *PDGFRA* expression (Fig. 6c and Supplementary Fig. 10e). Exon-level expression aligned with the fusion breakpoints. We further validated the fusion in both samples using RT-PCR (Fig. 6d).

## Discussion

Solid tumors are rare in children; the rarity poses a significant challenge for building resources at scale. Preserving tumor tissue in rodents is essential for preclinical and mechanistic studies and for resource sharing. Here, we have built a resource of 68 subcutaneous xenografts derived from pediatric solid tumors, including several very rare cancer types. All the PDXs have been molecularly characterized, and the tissue materials are ready to be distributed upon request.

With this resource, we determined conservation of mutations, SCNA, and expression profiles in PDXs. We found that early-passage PDXs faithfully retain expression profiles of the PT, suggesting gene expression is tumor-intrinsic. The conservation of gene expression is not dependent on model mutational similarity; thus, expression can be a more robust tool to transfer preclinical insights from PDXs to PTs. We

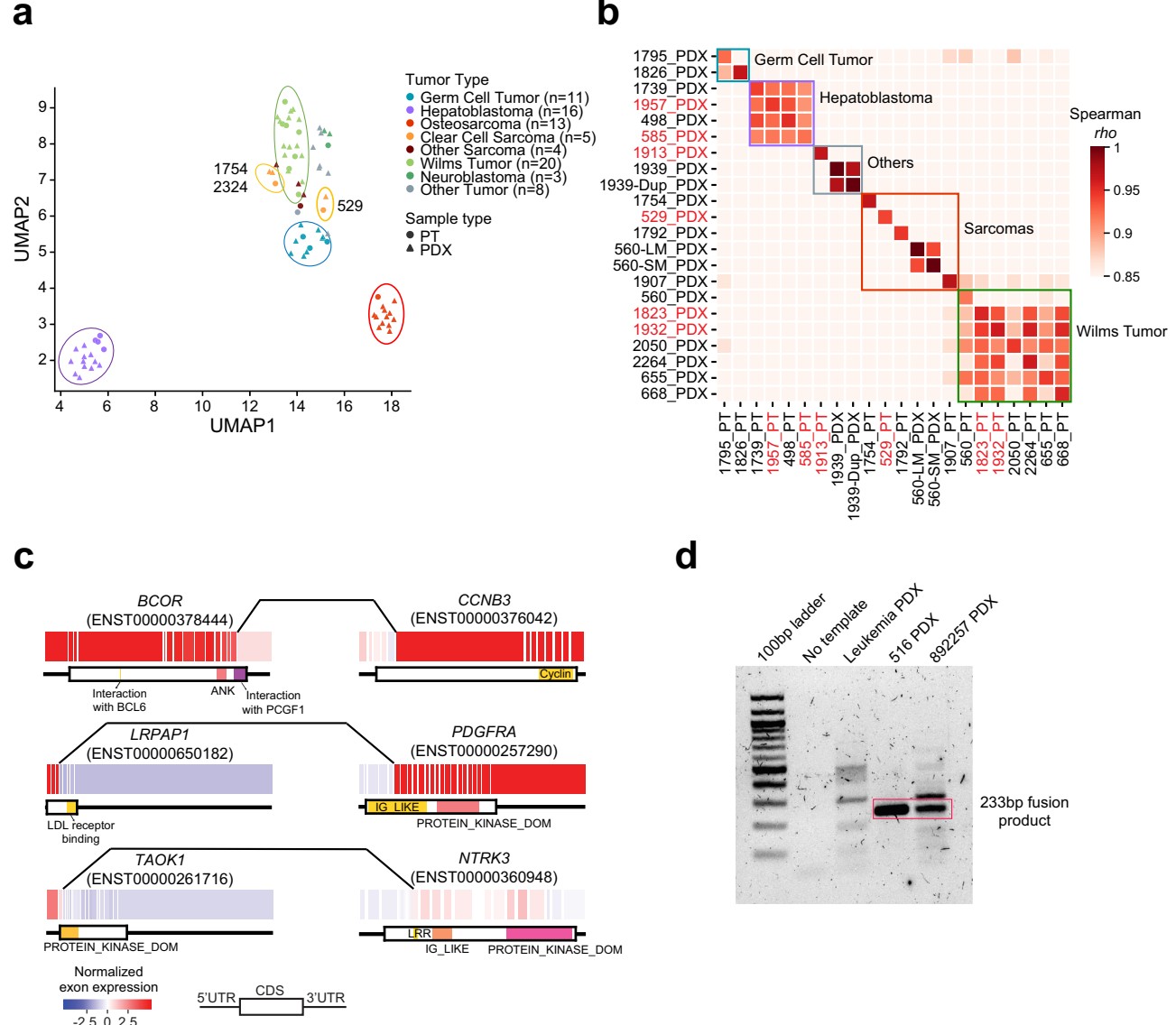

**Fig. 6 | Transcriptomic similarity and gene fusion. a** UMAP visualization for samples with expression data. Each dot represents a sample (circle, PT; triangle, PDX), and the color reflects tumor type. **b** Expression similarity of paired samples. The correlation coefficient was calculated using Spearman's correlation. Samples labeled in red are from group 2. **c** Example gene fusions. Each box represents an exon, and lines connecting the two genes indicate fusion breakpoints. Protein domains are shown below the exon plot. **d** RT-PCR validation of LRPAP1-PDGFRA fusion transcript. The red rectangle indicates LRPAP1-PDGFRA fusion bands. A leukemia sample was included as negative control. This experiment was repeated twice. Source data are provided as a Source data file.

observed that PDXs generally conserve the SCNA profiles of the PT. The conservation of SCNAs was also observed in adult cancer PDXs[14]. These observations are consistent with results from single cell sequencing that suggested most SCNAs are early events during transformation[49]. Further insights in SCNA stability can be gleaned through its characterization over serial PDX passaging[19,50].

We found significant mutation disparity in ~30% of the PDXs, and this disparity was associated with high genetic heterogeneity of the PT. Thus, more heterogenous tumors tend to generate genetically more different PDXs. The association could result from sampling bias, where a tumor block distinct from the PT seeds the PDX. Alternatively, engraftment disrupts the clonal equilibrium of the cancer ecosystem and a subclone outcompetes other clones to become the dominant clone in the PDX. Both mechanisms are possible, but sampling bias is unlikely the major force as it cannot explain the longer engraftment time and higher proliferation of the PDXs that we observe in nearly every genetically disparate pair. Supporting this concept, additional

PDXs established from distinct tumor tissue of patients 1913 and 1932, where significant mutation disparity between the PT and the original PDX was observed, demonstrated similar subclonal expansion.

Cancer heterogeneity fosters evolution[51], but evolution is driven by environmental changes. Implantation of cancer cells in immuno-compromised mice removes antitumor immunity for cancer cells, thus potentially allowing previously constrained, more immunogenic clones to grow. In support of this idea, we show that genetically disparate PDXs express significantly more clonal neoantigens than their matched PTs. For example, in patient 1957, we found two shared mutations between the PT and PDX. Of the 39 PT-specific mutations, none were predicted to encode a clonal neoantigen. In contrast, 10 clonal neoantigens were predicted out of the 32 PDX-specific mutations. The bulk of these PDX-specific neoantigens were unlikely acquired during PDX production. In mutational signature analysis, when chemo-related mutational signatures were identified in PTs, the same signatures were also identified in the PDXs with all PDX

mutations or PDX specific mutations, even though the mice were never treated. Moreover, seven PDX specific clonal mutations from patient 1913 were also found in the two additional PDXs that were established from distinct patient tumor tissue. This observation provides evidence that these mutations preexist in the primary patient tumor, because it is virtually impossible for PDXs grown in different mice to acquire several identical mutations. The preexistence of PDX-specific mutation is also consistent with the low mutation rate of childhood cancers.

Recently, PDX co-clinical trials have been proposed to guide therapy at the time of tumor relapse[52]. Our data suggest such co-trials could misinform treatment when the tumor is highly heterogeneous. In addition, we show in some patients, the immune system, particularly the innate immune system, might have suppressed the more aggressive cancer subclones. Thus, leveraging this antitumor immunity in combination with surgical resection may benefit these patients.

In summary, we build a PDX resource for pediatric solid cancer research and describe evidence that the interplay between intratumor heterogeneity and immune constraints on tumor evolution may underlie the genetic disparity between PTs and PDXs. More studies with larger cohorts are warranted to further validate and extend this finding, including in adult cancers.

## Methods

### Sample collection and patient consent

This study complies with all relevant ethical regulations. It was approved by the Institutional Review Board (IRB) and the Institutional Animal Care and Use Committees (IACUC, protocol #15015) of University of Texas Southwestern Medical Center (UTSW), Dallas, TX and UT Health San Antonio (UTHSA), TX. The human aspect of this study was deemed minimal risk by the approving IRB. No specific ethics review was therefore required. Individuals with identified solid tumors, both benign and malignant, were approached and offered enrollment on an institutional, IRB approved biorepository prior to standard of care surgical procedures. For patients less than 18 years of age, parents or legally authorized representatives provided consent. Assent of the patient was required for those participants 10–17 years of age. All anatomic sites of disease were eligible and individuals were considered eligible if they were under 30 years of age at the time of consent. Patient race and ethnicity was self-reported. Biorepository consent included collection of medical waste for research, including the tumor utilized here, and consented for the generation of patient derived xenografts. Patients also consented to the collection of germline DNA, collection of basic demographic information and outcome data as part of the biorepository. Tissue from the surgical procedure which was considered excess or not necessary for diagnosis was collected, de-identified, and prepared for shipment and/or injection for development of patient derived xenografts. Tissue samples were kept at 4 °C until prepared for shipment and/or injection. All study procedures were completed after initial consent was obtained. Patient sex was not considered in the study design.

### Establishment of solid tumor PDX model

Patient-derived Xenografts (PDX) were generated from childhood cancer patients as described earlier[27] with minor modifications described here. Subcutaneous human xenografts from patient derived tumors were generated in a highly immunodeficient NSG (NOD.Cg-Prkdc IL2-Rgnull/Szj) mouse model (Jackson Laboratories, Bar Harbor, ME, USA).

Tumor specimens were collected immediately after biopsy/surgery in antibiotic (2% penicillin/streptomycin) containing M199 medium. The specimens were transported same day to GCCRI from University Hospital or Methodist Hospital, San Antonio or shipped overnight from other institutions (UTSW and APEC14B1 project hospitals) to GCCRI using cold shipping containers and transplantation was performed the same day specimens were received.

Female NSG mice were received at 6–8 weeks of age and allowed a 7–10 day acclimation period. Animal cages are changed out once a week or every other week depending on housing type, e.g., micro-isolator or individually ventilated cages. The cages are maintained in animal rooms equipped to provide 10–15 air changes per hour. The room temperature is maintained at 21–26 °C, relative humidity between 30 and 70% with a 14:10 day:night light cycle. Transplantation was performed once mice gained an average body weight of 20 g. Hair was removed at the site of incision (above the base of tail over the spine). The transplantation was performed under a biological laminar flow cabinet, under anesthesia (in an induction chamber with the flow of 5% isoflurane at 4 L of oxygen per minute) until mice were unresponsive to a toe pinch.

The tumor fragments (made about 2 × 2 mm) were kept in fresh medium until the mouse was ready for transplantation. The site was swabbed with 70% ethanol, an incision was made (approx. 4 mm) and a pocket was created under the skin using scissors. Using forceps, a tumor piece was dipped briefly into Matrigel (supplemented with VEGF, 100 ng/ml, to enhance angiogenesis) and placed inside the pocket followed by irrigation with a drop of penicillin/streptomycin and the incision was closed by using a small drop of tissue glue (Vetbond).

### Collection of PDX passages as viables and snaps

Mice were monitored for xenograft growth and healing of the incision. When tumors reached a size of about 1 × 1 cm, mice were terminated and tumors were collected as viables (in 7.5% DMSO/50% FBS) and kept frozen in a liquid nitrogen vapor tank and also snap frozen in liquid nitrogen to get Snaps for genomic analysis and preserved at −80 °C. Tumors were also further transplanted into additional mice (typically 5 mice) as donors for passaging of PDX. The maximal tumor size permitted by IACUC (800–1600 mm³) was not exceeded.

### DNA and RNA sequencing

Genomic DNA was extracted with DNeasy Blood & Tissue Kit (QIAGEN). KAPA HyperPrep kit was utilized to construct DNA libraries for whole genome sequencing and whole-exome sequencing. Approximately 250–500 ng genomic DNA were sheared with Covaris S220 Ultra Sonicator to the average of 200–400 bp fragments for DNA-seq library preparation. Then one proportion of DNA-seq libraries was quantified and pooled together for whole genome sequencing using 150 bp paired end sequencing; other proportion of DNA-seq libraries (around 250 ng) were quantified and pooled to go through two rounds of hybridization to enrich the DNA fragments of exome regions by using IDT xGen Exome Research Panel (V1 and V2). The final WES library was amplified, quantified, and loaded for 100 bp paired end sequencing at UTHSA Genome Sequencing Facility. On average, WES was sequenced to 300× and low-pass WGS was sequenced to 4×.

RNA was isolated using RNeasy Mini Kit (QIAGEN). The quality of Total RNA was checked by Agilent Fragment Analyzer (Agilent Technologies, Santa Clara, CA), and only high-quality RNA samples (RQN > 7) were used for mRNA-seq library preparation and sequencing. Following the Illumina TruSeq stranded mRNA sample preparation guide, we used approximately 500 ng Total RNA for RNA-seq library preparation. After RNA-seq libraries were quantified, they were pooled and subsequently loaded for 100 bp paired read sequencing run on the Illumina HiSeq 3000 platform. An average of 80 million reads were obtained per sample.

### Target enrichment and deep sequencing

Based on the mutation calling result from WES data, we designed the probes for unique somatic point mutations found in 585_PT and 585_PDX. Approximate 100 ng whole genome DNA was used for DNA-seq library preparation with Twist Library Preparation EF2.0 Enzymatic

Fragmentation Kit (104206, Twist bioscience, San Francisco, CA). The whole genome DNA was sheared by enzymatic fragmentation and the fragmentation time has been optimized to generate the mode fragment length about 200–300 bps. Then following end repair & A-tailing and adapter ligation, SPRI beads size selection was used to ensure the library insert size uniform. After PCR amplification, the final DNA-seq library is cleaned up with SPRI beads and quantified with Qubit and Fragment Analyzer. Then target capture libraries were prepared by following Twist Custom Panel Hybridization Capture of DNA libraries protocol (Twist bioscience, San Francisco, CA), and final libraries were quantified with Qubit and Fragment Analyzer. Final libraries were then loaded on NovaSeq 6000 System with 150 bp paired end sequencing. After the sequencing run, sample demultiplexing is performed to generate FASTQ files× for each sample. The average depth of sequencing is 5000–7000×.

Target-capture-based deep sequencing was often used in PT/PDX comparisons. While it can improve detection sensitivity of mutations with low allele fractions, it does not and should not be used to identify mutations that are not detected by whole exome or genome sequencing data. The results from deep sequencing data should also be interpreted with caution. Whereas the detection of a PDX-specific mutation in the PT by deep sequencing suggests pre-existence of the mutation in the parental patient tumor, the absence of the mutation in the PT does not prove the mutation is acquired de novo by the PDX because the PT sample, where the deep sequencing is done, is not the tissue origin of the PDX. Sequencing coverage in this context should be also considered for data interpretation.

### Sequencing data preprocessing and quality control

Trim Galore[53] (v0.6.7) was applied to raw sequencing data to remove the adapter and poor-quality reads. BWA-MEM[54] (v0.7.17) and STAR[55] (v2.7.9a) were used to align DNA and RNA sequencing data to the reference genome. To remove mouse-derived reads in PDXs, we mapped the sequencing data to the human (GRCh38, GENCODE v29) and mouse (GRCm38, GENCODE vM19) reference genomes from GENCODE[56]. Disambiguate[57] (v1.0) was then employed on the BAM files to remove mouse reads. Notably, for RNA sequencing data, we converted the BAM file of human reads to FASTQ format using Samtools[58] (v1.14), so that we can merge them with the unmapped reads for gene fusion detection. GATK[59] best practice workflow was used to deduplicate and recalibrate the aligned BAM files for DNA sequencing data.

PDXs with a mouse contamination rate >50% were excluded from further analyses. Samples with this high contamination rate included one RNA-seq (1853), three WES (1796, 1853, 512) and two WGS PDXs (1796, 1853). We further excluded two WES PDX samples (560-SM, 707) that had low coverage after mouse read removal. NGSCheckMate[60] (v1.0.0) was applied to ensure matching between PTs and PDXs using both DNA and RNA sequencing data. In addition, RNA-seqQC[61] (v2.3.5) and Samtools were applied to RNA and DNA sequencing data to assess mapping quality.

### Mutational analysis

MuTect2 (GATK v4.2.3.0), VarScan (v2.4.4), Strelka (v2.9.10) and Pindel (v0.2.5b9)[62–65] were utilized to identify somatic mutations and indels from the WES data. To filter false positives, DKFZ's bias filtering (https://github.com/DKFZ-ODCF/DKFZBiasFilter) was used to filter mutations with strand bias or bias toward PCR template strand. We used fpfilter.pl (https://sourceforge.net/projects/varscan/files/scripts) to remove false positives from VarScan output. We excluded mutations in intergenic, intron, or outside capture regions. To remove potential germline variants, we annotated the remaining mutations using population databases (including 1000 genome phase 3, ESP6500, non-TCGA ExAC and gnomAD 3.0)[66], and only kept variants with MAF < 0.001. We further

removed variants that were found in either TCGA panel of normal or the panel of normal generated from this dataset. Then, we filtered out multiallelic mutations and double/triple nucleotide polymorphisms (DNP and TNP), and only included insertions or deletions shorter than 50 bp. Next, we required mutations to have at the minimum tumor depth ≥14, normal depth ≥8, tumor VAF ≥ 0.05, normal VAF ≤ 0.01 and tumor mutant allele reads ≥ 4. High confidence somatic mutations were identified as those that were called by at least two callers. Notably, for PT-PDX paired sample, if a mutation was detected only in one sample, we rescued the mutation in its paired sample if this mutation was found by any of the tools in the raw outputs. To test if this rescue strategy would miss any mutations, we used bam-readcount[67] to examine all the 294 PT or PDX private mutations in the matched sample. This supervised approach only found 2 PDX private mutation (<1%) with very low VAFs (0.024 and 0.014) in the matched PT, and found no evidence of PT private mutations in the matched PDX.

Several adaptions were made for tumors without a matched normal. For these tumors, we used MuTect2 tumor-only mode, and the 40 normal samples in our dataset were used as the panel of normal. Pindel was not used because it requires the matched normal. Mutations were considered high confidence if they were detected by all three tools. After rescuing mutations in paired samples, we used SGZ[68] to predict if a variant was somatic or germline. We excluded the predicted germline or probable germline variants unless the variant is cataloged by the COSMIC database or located in cancer genes[69], and kept those predicted as somatic or likely somatic. If an identified germline variant was found in one of PT-PDX paired samples, we removed it in both samples.

To test if the multi-caller approach would miss hotspot mutations, we compared mutations downloaded from the MSKCC hotspot database with the mutations that have been filtered out. Of the 3554 somatic mutations that were called by only one caller, only one point mutation (*NUP93*, E14V) and two indels of *CTNNB1* were documented in the MSKCC hotspot database. Thus, the number of missed mutations is negligible. Since *CTNNB1* harbors frequent indels in hepatoblastoma, we applied a supervised approach to identify them, see 'other mutation related analyses'.

To validate the mutation calling, we examined mutations called from WES in RNAseq and low pass WGS sequencing data. Of the 388 mutation sites with coverage ≥10 in either RNAseq or low pass WGS, 356 mutations showed at least one read covering the mutant allele, resulting in a validation rate of 92%.

We used oncoKB-annotator to determine functional consequences of the mutations. In total we identified 46 oncogenic or likely oncogenic mutations, 30 of which were found in PT/PDX pairs. Among the 30, 28 were shared between matched PTs and PDXs.

### Tumor purity and ploidy prediction

For samples with a paired normal, tumor purity and ploidy were estimated using Sequenza[70] (v3.0.0). For samples without a paired normal, tumor purity and ploidy were estimated using PureCN[71] based on CNV segmentation by CNVkit[72] (v0.9.9). Tumor purity was also estimated from RNAseq data using ESTIMATE[73].

### Consensus clonality analysis

We applied four methods to characterize mutation clonality. The first method was described by McGranahan et al.[74]. Using the method, we estimated cancer cell fraction (CCF) for each mutation and classified mutations as clonal if their CCF confidence interval overlaps 1, or as subclonal if otherwise. We additionally used PyClone-VI (v0.1.1), CliP (v1.2.1) and Ccube (v1.0)[75–77]. These methods cluster mutations and then estimate the corresponding CCF of each cluster. Based on the outputs of these methods, we identified clonal and subclonal mutations by the following criteria:

- If only one cluster was found, all mutations within the cluster were regarded clonal.
- If more than one clusters were found, mutations of the cluster with the highest mean CCF were regarded clonal. For the remaining clusters, if the mean CCF was larger than 0.9, mutations within those clusters were also regarded clonal. This relaxed criterion was used to accommodate uncertainties associated with CCF estimates. The others were regarded subclonal.

The consensus mutational clonality was built on votes from these four approaches. A mutation was identified as consensus clonal if it was identified as such by at least two methods, and similarly for subclonal mutations. Other mutations were treated as ambiguous. Mutations without the needed copy number information to infer clonality were not classified. The clonality flow between PTs and PDXs was plotted by R package ggalluvial[78] using the consensus calls. We also manually examined VAFs and copy number status of those mutations between each paired PT and PDX to corroborate the evolutionary pattern.

### Neoantigen prediction

The 4-digit HLA typing of each sample was predicted using Optitype[79] (v1.3.5). Based on the non-synonymous mutations and HLA typing, pVACseq[80] (pVACtools suite v3.0.0) was applied to identify peptides of 8–11 amino acids. The peptide binding affinity to MHC was predicted using NetMHCpan (v4.1), PickPocket (v1.1), SMM (v1.0) and SMMPMBEC (v1.0)[81–84]. Neoantigens were identified as peptides with best MT IC50 ≤ 500 nM. We in total identified 305 neoantigens in PT-PDX paired samples, of which 239 were clonal.

### Mutation signature analysis

The known mutational signature matrix (v3.2, GCRh38) was downloaded from COSMIC[32]. We determined mutation signatures using the R package deconstructSigs[85] (v1.8.0). For signature analysis, we required a sample to have at least 20 somatic mutations. For visualization, signatures with weight less 0.25 across all samples were excluded. We did not exclude any signatures during deconvolution, and no signature scaling was applied.

### Other mutation-related analyses

To calculate microsatellite unstable (MSI) scores, we used MSIsensor[86] for tumors with a matched control and MSIsensor2[86] for tumors without a matched control. To identify large in-frame *CTNNB1* deletions in hepatoblastoma, we used MANTA[87] (v1.6.0) to identify the structural breakpoints in exon 3 or 4 of *CTNNB1*, following a previous study[29]. By this approach, we further added 4 *CTNNB1* deletion back to our mutation calling result. We applied Telseq[88] (v0.0.1) to both WGS and WES data to estimate the average telomere length of each sample.

### Somatic copy number alterations (SCNA)

Copy number (CN) segmentation was calculated from WGS data using CNVkit[72] (v0.9.9) with default parameters. For tumor-only cases, we generated the copy number reference from 40 normal samples. Absolute copy number was estimated using PureCN[71] (v2.2.0) best practice pipeline with segmentation generated by CNVkit. GISTIC2[89] was used to call recurrent peaks of all samples with parameter "-conf 0.99 -armpeel 1 -ta 0.3 -td 0.3."

To compare the copy number profiles between PTs and PDXs, we first divided the genome into 1 Mb window bins using BEDTools[90] (v2.30.0). After removing segments located in centromeres or telomeres, we calculated the weighted mean of each bin across all samples. Copy number similarity was quantified by Pearson correlation based on the weighted mean matrix. Similarly, for CN similarity of cancer genes, the similarity was calculated using Pearson correlation based on the weighted mean of each gene. The cancer gene list was downloaded from Cancer Gene Census[69]. Here, we only used genes identified as oncogenes or tumor suppressors (listed in Supplementary Data "Cancer genes").

Focal copy number variations were identified with segment length <50% of the chromosome arm and with copy number ratio >0.3 or <−0.3. Given the potential inconsistency of focal SCNAs breakpoints in PTs and PDXs, if a focal event was amplified or deleted in both PT and PDX samples and the breakpoints of these two segments were within +/−10 kb range, this event was regarded as a shared focal event. Finally, for paired PT or PDX samples, we also rescued focal SCNAs with breakpoints within +/−10 kb range and with copy number ratio >0.1 (for amplification) or <−0.1 (for deletion) in its paired sample. We in total rescued 65 focal SCNAs (22% of all focal events).

The genomic instability (GI) of each chromosome arm was calculated as the proportion of gains (>0.3) or losses (<−0.3). The total CIN of each sample was defined as the mean of arm-level GIs.

### Gene expression analysis

Kallisto[91] (v0.46.0) was used to calculate transcript per million (TPM). For unsupervised clustering analysis, UMAP was used based on the top 1500 variable genes identified by median absolute deviation of TPM after removing immune-related, mitochondrial, and ribosomal genes. The immune-related genes were identified as those whose gene expression was positively correlated with ESTIMATE immune score ($p < 0.05$, Pearson correlation). We removed immune genes because these genes are either low or absent in our PDXs. Including them would bias the clustering of PDXs and PTs. We used the UMAP function implemented in the python package umap-learn (v0.5.1) with parameters "n_neighbors=15, min_dist=0.15". The similarity between PTs and PDXs was calculated with Spearman's correlation after excluding immune-related, mitochondrial, and ribosomal genes. HTseq-count[92] (v0.13.5) was used to generate exon expression of each gene with parameters "-s no -t exon -m union --nonunique all". RSEM[93] was used to estimate raw read counts.

### Gene fusion identification

To detect fusions in PDXs, we merged unmapped reads with human-only reads from Disambiguate. The unmapped reads may contain junction spanning reads. Both STAR-Fusion[94] (v1.10.0) and PRADA2[4] were applied to detect gene fusions. After removing fusions that were observed in normal samples, i.e., those annotated as "GTEx_recurrent", we obtained 916 fusions with STAR-Fusion. With PRADA2, we obtained 237 fusions. Fusions identified by both methods were regarded as high confidence. For PT-PDX or PDX-PDX paired samples, we rescued PT or PDX only fusions in its paired sample from the raw fusion pool of STAR-Fusion or PRADA2. We in total rescued 7 fusions (4.3% of all gene fusions). The functional consequence of fusion candidates (in-frame or out-of-frame) was predicted with PRADA2[4].

To validate the PDX fusion transcript LRPAP1–PDGFRA, we designed a pair of primers located on LRPAP1 and PDGFRA around the predicted fusion site: forward−5′ GCCAAGTATGGTCTGGACGG and reverse−5′ CGGGCAGCACATTCGTAATC, respectively. Product length was 233 bp. Total RNA was isolated from 30 mg of snap frozen PDX tissue using RNeasy mini kit (Qiagen, Cat#74004). Using One-step qRT-PCR kit (Invitrogen, Cat#11732-020) we performed one-step RT-PCR to amplify the predicted fusion gene junction from the same PDX tissue as was used for sequencing: 516_PDX & 892257_PDX. 50 ng of total RNA input was used for RT-PCR reaction. RT-PCR was performed in 50 μl reactions using 0.5 mM dNTPs, 3 mM MgSO4, 0.2 μM each primers and provided mix of SuperScript III RT/Platinum Taq. The RT-PCR reaction was carried out with the following program: 500 C for 30 min, followed by 950 C,

2 min and by 400 cycles of 950 C, 15 s, 550 C, 30 s and 680 C, 1 min. RT-PCR products were analyzed by agarose gel electrophoresis (2%). The result was visualized with SYBR safe DNA gel stain (ThermoFisher Sc, Cat#S33100).

## Other analyses

The ssGSEA and GSEA analysis was done using the python package GSEApy (v0.10.4)[95]. MsigDB C2 collection[96] (c2.all.v2022.1.Hs.symbols.gmt) was used in GSEA analysis to find the significantly differential pathways between Group1 and Group2 patient tumors. To obtain cell proliferation activity, we applied ssGSEA to cell proliferation signatures, including Benporath_Proliferation, REACTOME_Cell_Cycle, and KEGG_Cell_Cycle. We also applied ssGSEA to patient tumors to estimate activity of immune-related gene signatures. The gene signatures were collected from previous studies[97,98]. Besides, we used TIMER2.0[99] to estimate abundance of immune cell infiltration in PT samples.

## Statistics and reproducibility

No statistical method was used to predetermine sample size. PDXs were excluded from the analyses when high mouse contamination was detected in the genomic data. The experiments were not randomized.

## Data availability

The raw sequencing data generated in this study have been deposited in the European Genome-Phenome Archive database under accession code EGAS00001006710 [https://ega-archive.org/datasets/EGAD00001009863]. The processed genomic data are available at synapse (Synapse ID: syn35811916). PDX clinical information and request forms can be found at the pediatric solid tumor PDX portal [https://pstPDX.streamlit.app]. The processed data generated in this study are provided in the Supplementary Information/Source data file. For the raw sequencing data that are under controlled access on EGA, access information including data access agreement and conditions of data release is provided on the portal site. Data access requests can also be sent to cprit_tpct@uthscsa.edu. Data access will be granted as soon as data requests are approved by the data oversight committee. No restriction is placed on how long the data will be made available for; however, data availability is bound by the scope and duration of the research projects described in the data access agreement. Source data are provided with this paper.

## Code availability

The codes used for sequencing data analysis are available on Github at https://github.com/fnhe/PediatricSolidTumorPDX[100].

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

## Acknowledgements

We thank Kathryn Bondra, Fuyang Li, Vanessa DelPozo, Samson Ghilu, and Edward Favors for technical assistance with sample preparation and animal work. We thank UT Health Information Management Systems for IT support. This work was supported by CPRIT (RP160716, RP220599 to P.J.H., RP180319 to S.X.S., RR170055 to Z.S.). Z.L. is supported by NIH NCI R50CA265339. The Sequencing data used in the study were generated at The Greehey Children's Cancer Research Institute (GCCRI) Genome Sequencing Facility (GSF). GSF is a Mays Cancer Center Next Generation Sequencing Shared Resource (NGSSR) and is supported by NIH-NCI P30 CA054174, NIH Shared Instrument grants S10OD021805 and S10OD030311 (Z.L.), and CPRIT Core Facility Awards RP160732 and RP220662 (Y.C.).

## Author contributions

Project supervision and funding acquisition: P.J.H., S.Z., R.K., S.X.S., G.E.T. and X.Y. PDX production and sample preparation: A.M.B., R.K., A.R., T.H. Fusion validation: A.R. Genomic data analysis and portal development: F.H., S.Z., Y.C., L.X., X.W., D.K., S.H.C. Patient tumor and clinical data: L.J.K., G.T., L.-N.P.P., E.B., T.H., A.-M.L., A.G., A.S., S.S., C.A. Genomic data production: Z.L., Y.Z., D.G., K.W. Manuscript writing, with input from all authors: S.Z., F.H., P.J.H., R.K.

## Competing interests

L.J.K. consults for Alexion Pharmaceuticals without a fee. The other authors declare no competing interests.

## Additional information

[1]Greehey Children's Cancer Research Institute, University of Texas Health Science Center, San Antonio, TX, USA. [2]Department of Population Health Sciences, University of Texas Health Science Center, San Antonio, TX, USA. [3]Department of Pediatrics, Division of Hematology/Oncology, University of Texas Southwestern Medical Center, Dallas, TX, USA. [4]Harold C. Simmons Comprehensive Cancer Center, University of Texas Southwestern Medical Center, Dallas, TX, USA. [5]Gill Center for Cancer and Blood Disorders, Children's Health Children's Medical Center, Dallas, TX, USA. [6]Department of Biochemistry and Structural Biology, University of Texas Health Science Center, San Antonio, TX, USA. [7]Department of Pediatrics, University of Texas Health Science Center, San Antonio, TX, USA. [8]Mays Cancer Center, University of Texas Health Science Center, San Antonio, TX, USA. [9]Department of Molecular Medicine, University of Texas Health Science Center, San Antonio, TX, USA. [10]Quantitative Biomedical Research Center, Peter O'Donnell Jr. School of Public Health, University of Texas Southwestern Medical Center, Dallas, TX, USA. [11]Department of Bioinformatics, University of Texas Southwestern Medical Center, Dallas, TX, USA. [12]These authors contributed equally: Funan He, Abhik M. Bandyopadhyay. ✉e-mail: kurmasheva@uthscsa.edu; zhengs3@uthscsa.edu

