## [Peer Review File · Nature Communications]

REVIEWERS' COMMENTS

Reviewer #1 (Remarks to the Author): expertise in multi-omics integration methods

Reviewers: Jo Lynne Rokita, Ph.D. and Ryan J. Corbett, Ph.D.

He et al. report extensive molecular profiling of patient-derived xenografts (PDXs) and matched primary tumor (PT) samples derived from multiple pediatric cancer types, and compare mutational landscapes, clonality, and gene expression between matched samples. The authors identify variation in mutational similarities across PDX-PT pairs, and present two distinct patterns of engraftment evolutionary patterns characterized by clone retention (group 1) and subclonal expansion/branch seeding (groups 2 & 3). The authors present multiple lines of evidence that the unique evolutionary profiles of group 2 & 3 PDXs is likely driven by an immunogenic subclone in the PT that rapidly expands in the absence of immune surveillance. The authors further show that, unlike SNVs and indels, somatic copy number alterations and gene expression are highly similar between PDXs and matched PTs. Some of the results presented in this manuscript are available to view interactively through an online portal. Overall, this manuscript provides novel insight into observed disparities between PDXs and PTs that are driven in part by PT heterogeneity and we are enthusiastic about the work being published for the pediatric oncology community. However, there are concerns regarding the reproducibility of this work that should be addressed before full manuscript consideration.

- General

- o This reviewer suggests the use of "mutation" for somatic alterations and "variant" for germline alterations throughout the manuscript in order to alleviate confusion.
- o While most of the analyses in this manuscript are restricted to matched PDX and PT samples from the same patients, PDX without a matched PT sample, and vice versa, are also included in some of the comparisons made between PDX and PT samples (in Supp figures 3 and 4, for example). It would be helpful for the readers if there was clear designation for analyses using paired samples versus all samples.
- o Lines 130-133. Is it known what chemotherapy treatments these 10 patients received, and whether these are associated with the specific mutational signatures found? For example, SBS31 and SBS35 are associated with platinum chemotherapy treatment and SBS87 thiopurine treatment. Also, were other treatment-related signatures observed (SBS25, 32, 86)?
- o Figure 2a is slightly hard to read – a suggestion would be to remove the box colors making a transparent box showing only the jitter points and shapes.
- o Figure 2b. It is said in the results text that 7/8 high mutation samples have therapy-related mutational signatures, but it looks as though all samples have some non-zero value shading in cells for at least one therapy mut sig. If there is some exposure threshold that signatures must meet to be considered "present" in a patient, this needs to be made clear in the heatmap (either by zeroing-out sigs that don't meet threshold, or an asterisk-like indication in cells that do meet threshold).
- o It is interesting that there are some PDX-specific oncogene or tumor suppressor gene mutations (Supplemental Figure 4d-e). Were they all predicted to be oncogenic/likely oncogenic or are they VUS (eg OncoKB)? Perhaps an updated figure which only includes oncogenic/likely oncogenic (eg predicted by OncoKB) would be more functionally relevant if these are mixed. Are these oncogene clonal mutations present at subclonal levels in the PT tumor even below variant calling criteria in the PT tumor (ie, did the authors do a deep dive outside of 585, even in the absence of additional deep sequencing)? Likewise, the authors can investigate this from the other direction, but the unexpected finding would be a novel oncogene mutation in the PDX which never existed in the PT tumor and what is the mechanism of this?
- o The "clone sweeping" examples vary from expansion of one subclone or many subclones, so the text can be that this is defined as at least one subclone expanding from PT to PDX.
- o Figure 3g would likely be easier to interpret if it were shown as a box/dot/violin plot.
- o Lines 210-211: a correlation test should be performed to show statistical evidence for PT genetic heterogeneity being inversely associated with mutational similarity.
- o Lines 306-307: Should this say "hepatoblastoma and Wilms tumor samples showed significant

intra-lineage correlations" (remove "Except for sarcoma").

o The observation that more mutationally diverse PT tumors yield PDX with differing genetics makes sense, but it does appear that 2/3 branch seeding models (1957 and 1823) were derived from PTs treated with chemotherapy, so this may not be unexpected. Additionally, can this be a tumor+chemotherapy-specific phenomenon? It appears that 2 hepatoblastoma models pre-therapy nicely recapitulated PT tumors, but the two with chemotherapy did not. In other cancer types, there is not a similar trend, albeit overall numbers are low. Were the retinoblastoma PTs treated with a different agent?

- Molecular Methods

o The kits or protocols used for DNA and RNA extraction are not described in the Methods text.

- Computational Methods

o "NGSCheckMate9 (v1.0.0) was applied to both DNA and RNA sequencing data to make sure matched samples were from the same patient." This reads as if PDX DNA and RNA were checked for matching, but was this also done between PDX and patient to ensure PDX/patient matches?

o The text states that "The BAM file of human reads was converted to FASTQ format using Samtools7 (v1.14) for further analyses". This seems incorrect; was there another reason for re-generating fastq files with human-only reads?

o "High confidence somatic mutations were identified as those that were called by at least two callers." Were there any hotspot mutations missed by two callers, or were these checked, for example using the MSKCC hotspot database: <https://www.cancerhotspots.org/#/home>

o What was the genetic sex makeup of the 40 PON samples? Is it balanced to ensure quality calls in XY chromosomes? What was the accuracy of the calls using 3/3 algorithms in tumor only samples? Did this result in a loss of any oncogenic calls?

o Mutational signature analysis: it is unclear what is meant by "We excluded ...signatures with the highest weight < 0.25 for further analysis." Were mean signature weights recalculated using only the remaining signatures? (ie- if when using "all signatures ABCDEF", Signature E had <0.25 in all samples, then was deconstructSigs rerun for only ABCDF signatures?) Additionally, it is common to filter out any signatures in patients with exposure weight < 0.06 (i.e., set them to zero). The results text and corresponding figure for this analysis make it unclear if this was performed.

o "The cancer gene list was downloaded from Cancer Gene Census17." The authors can include this gene list as a supplemental table since this can change over time.

o It is stated in the Methods that differential gene and pathway expression analyses were performed, but these results are not presented (gene) or shown in reduced form in Supp. Table 5 (enriched pathways). These results should either be presented in full in a supp. table AND discussed in the text, or the methods should not be included.

o For differential expression analysis, the authors state that a multifactor analysis was performed but do not indicate if any additional covariates were included in the model (tumor type, treatment status, etc).

- Reproducibility

o The code used to generate the results has not been shared within the manuscript text. The authors should share all computational analyses used in this manuscript via Github or some other repository in order to enable reproducible analyses of these data.

Reviewer #2 (Remarks to the Author): expertise in solid tumour model development

In their manuscript titled "Genomic profiling of subcutaneous patient-derived xenografts reveals immune constraints on tumor evolution in childhood solid cancer," He et. Al. describe their genomic profiling of a small cohort of pediatric solid tumors (90). Their engraftment rates were similar to several previous publications. Only 68 tumors were genomically profiled with low pass WGS, WES and RNA-seq. Only 27 of those 68 had the matched patient tumor and 40 of the 68 had the germline. With these limited data, they report clonal enrichment as has been shown in multiple studies previously. They claim that the clonal selection may be due to anti-tumor immunity in the patient relative to the PDX. The evidence for this comes from the observation that in some PDXs, the minor clone in the patient predominates and those cells are more proliferative. Overall, this is a very small cohort with incomplete characterization and validation and the conclusions regarding anti-tumor immunity are not supported by the data.

Specific Comments:

A major limitation is the lack of patient tumor and germline for all of their cohort. The patient tumor and germline should be used with WGS, WES and RNA-seq to identify somatic lesions. The PDX and patient tumor should be used to assess clonal heterogeneity. Without all 3 samples (patient, germline, PDX) the impact of the study is minimal. Based on Fig. 1, there are only 22 with all 3 samples. Among those 22, 8 are Wilms, 2 are clear cell sarcoma, 1 is neuroblastoma, 3 are osteosarcoma, 4 are hepatoblastoma and 3 are germ cell tumor. It is very difficult to make any conclusion with such small numbers of tumor samples.

It is not clear why low pass WGS was performed rather than deeper WGS and there was no discussion of validation using custom capture and deep sequencing or at least cross validation of the WES and RNA-seq.

The mutation rate and mutation signature analysis were limited by sample size and there are much larger studies that have been published on patient tumors and PDXs with rigorous validation and deep sequencing.

It is interesting that they did perform capture enrichment and deep sequencing on a single patient tumor that had low PDX similarity. However, they did not perform the same analysis on the PDX. The proper way to do the study is to design capture probes for all somatic SNVs across the genome in each patient-PDX pair and then do deep sequencing on all SNVs on both samples. Using this approach, they can identify rare clones in both the patient tumor and the PDX.

A germline sample is required to serve as a reference to identify somatic mutations. It is not clear how they could conclude that mutations were clonal without doing the target enrichment and capture for all the SNVs across all the tumors. Subclonal SNVs may be lost in the shallow WGS and the exonic SNVs may have functional significance so interpreting clonal architecture in WES alone must take into account the rare SNVs and the possibility that they provide a selective growth advantage or disadvantage.

The clonal selection has been reported previously but there is no biologic or mechanistic data presented here to indicate that it is related to antitumor immunity rather than a stochastic event or some other mechanism. Indeed, if a minor clone overtakes the PDX, it is predicted to be more proliferative than in the patient but this does not prove anti-tumor immunity in the patient.

Reviewer #3 (Remarks to the Author): expertise in clone sweeping and genomics analysis

The authors generated patient-derived xenografts (PDXs) data from 65 pediatric tumors with various cancer types. Mutation similarity between the patient tumor (PT) and PDX was analyzed. The number of PDX datasets that were generated in this study is very large, and it will be a great resource. Also, the pattern of evolution in PDX is still largely unknown, so the findings are potentially useful. However, the results need to be more carefully interpreted.

1. In this study, only a single section of PT was sequenced, and similarities of mutation, copy number, and transcriptomics profiles between PT and PDX were analyzed. However, the observed difference can be due to sampling error, i.e., subclone/clone used for the xenograft was not sequenced in the PT data. Sequencing more sections from the PT may change the amount of observed similarity between PT and PDX. Ideally, sequencing more than one sample is necessary for this type of data analysis. Actually, sequencing multi-sections is very common in the analysis of tumor evolution, as sampling errors due to sequencing a single sample are a well-known problem.
2. Three evolutionary patterns were identified, i.e., clone retention, clone sweeping, and branch seeding. However, this classification can be also affected by sampling errors of clones in PT.
3. The low mutational similarity was concluded to be correlated with the high genetic heterogeneity of the PT without reporting clones within PT. Clones and clone phylogenies should be inferred, which is a common practice. Please note that clone prediction from a single tumor sample is not reliable, so caution is necessary.
4. Observed similarities of mutation, copy number, and transcriptomics profiles between PT and PDX are often not properly interpreted. For example, chemotherapy-related signatures were expected to be observed for PDXs when their corresponding PTs have those signatures, because

mutations that occurred in PTs are inherited in their PDXs. To test if mice without treatment accumulate mutations under the chemotherapy-related mutational processes, mutations that are unique only in PDXs should be compared with those from PTs. But, only a few datasets have a sufficient number of mutations that are unique to PDXs, so I think mutational processes in PDXs cannot be elucidated from these datasets.

5. The evolutionary patterns were concluded to correlate with PDX engraftment time. However, the number of unique mutations within PDX is expected to increase as the engraftment time. The mutation rate should be computed and compared.

6. Line178: The loss of PT clonal mutations can be also expected through copy number alteration (deletion). To reject this possibility copy number analysis is necessary.

7. Line 213: How were major clones identified?

RESPONSE TO REVIEWERS' COMMENTS

Reviewer #1 (Remarks to the Author): expertise in multi-omics integration methods

Reviewers: Jo Lynne Rokita, Ph.D. and Ryan J. Corbett, Ph.D.

He et al. report extensive molecular profiling of patient-derived xenografts (PDXs) and matched primary tumor (PT) samples derived from multiple pediatric cancer types, and compare mutational landscapes, clonality, and gene expression between matched samples. The authors identify variation in mutational similarities across PDX-PT pairs, and present two distinct patterns of engraftment evolutionary patterns characterized by clone retention (group 1) and subclonal expansion/branch seeding (groups 2 & 3). The authors present multiples lines of evidence that the unique evolutionary profiles of group 2 & 3 PDXs is likely driven by an immunogenic subclone in the PT that rapidly expands in the absence of immune surveillance. The authors further show that, unlike SNVs and indels, somatic copy number alterations and gene expression are highly similar between PDXs and matched PTs. Some of the results presented in this manuscript are available to view interactively through an online portal. Overall, this manuscript provides novel insight into observed disparities between PDXs and PTs that are driven in part by PT heterogeneity and we are enthusiastic about the work being published for the pediatric oncology community. However, there are concerns regarding the reproducibility of this work that should be addressed before full manuscript consideration.

We thank the reviewers for their enthusiasm, and for pointing out that our finding of distinct clonal evolution patterns and their association with immune surveillance is supported by “multiple lines of evidence.” Because this is our main message in this work, we are encouraged by the reviewers’ positive comments.

- General

1. This reviewer suggests the use of “mutation” for somatic alterations and “variant” for germline alterations throughout the manuscript in order to alleviate confusion.

Done.

2. While most of the analyses in this manuscript are restricted to matched PDX and PT samples from the same patients, PDX without a matched PT sample, and vice versa, are also included in some of the comparisons made between PDX and PT samples (in Supp figures 3 and 4, for example). It would be helpful for the readers if there was clear designation for analyses using paired samples versus all samples.

Done. We have clarified in figures whether “all samples” or “PT/PDX pairs” were used. See Supplementary Fig 3 for example.

3. Lines 130-133. Is it known what chemotherapy treatments these 10 patients received, and whether these are associated with the specific mutational signatures found? For example, SBS31 and SBS35 are associated with platinum chemotherapy treatment and SBS87 thiopurine treatment. Also, were other treatment-related signatures observed (SBS25, 32, 86)?

We thank the reviewers for raising this excellent question. Before we address this question, we note to the reviewers about two updates of our data. In our initial mutation signature analysis, we inadvertently missed four PDX samples (471, 466, 543, 1960), because either they or their

germlines were re-sequenced due to issues detected through our quality control steps. We later added their mutation data back to our MAF after the resequencing but forgot to include them in the mutation signature analysis. We apologize for the mishap. We now add these cases back. Adding them does not change the mutation signature results of any other samples.

Figure R1. Distribution of chemo-related mutational signatures in samples with at least 10 mutations. The asterisk represents signature weight larger than 0.25. In the revised manuscript, we only used samples with at least 20 mutations in mutational analysis (the first dashed vertical line on the left). Figure 2b has been updated accordingly.

The second update is related to SBS87. In COSMIC, SBS87 is associated with thiopurine treatment. We observed SBS87 in 2324, 1763, 1960, and to a lesser extent, in 2197. These patients received varying treatment regimens including AREN0321 (patient 2324), AREN0532 (patients 1763, 1960), and AEWS0031 (patient 2197), but none of the regimens contain thiopurine. We notice that samples demonstrating SBS87 in our cohort have low mutation counts (<15) (**Figure R1**). Signature deconvolution is generally less reliable with a small number of input mutations. A recent study on mutation signatures in pediatric cancer (Thatikonda et al. Nat Cancer, 2023; PMID: 36702933) applied a threshold of 20 mutations as the minimum. In the revision, we adopted their threshold and included only samples with at least 20 mutations. This change led to removal of SBS87 in our data. Though we now have less samples in this analysis, we feel more confident with the results.

In the new results (also the initial data), we found robust evidence of platinum drug associated signatures SBS31 and SBS35 in samples derived from patients 1792, 585, 1925, 560, 1957, and 543. Except for 560 that does not have treatment information, all the other 5 patients received a platinum-based drug (cisplatin). SBS86 was found in PDX_471. It is currently unknown what chemo-drugs can cause SBS86. Patient 471 received unspecified chemo-drugs before tumor collection. Thus, the signatures in these patients are supported by their treatment history.

We did not observe signatures SBS25 and SBS32 in our cohort. This is not unexpected. SBS25 was identified in Hodgkin’s lymphoma, which is not present in our cohort. SBS32 is associated with azathioprine. The signature is rarely seen in cancer; in the ICGC study (Alexandrov et al. Nature, 2020. PMID: 32025018), SBS32 was only seen in a very small fraction of biliary cancer and MDS, both of which are not present in our cohort.

We think the change we have made cleans the data and further sharpens our main message from this figure, that PDXs carry chemotherapy related signatures despite no exposure to chemo-

agents. In the revision, we have updated Figure 2b and Method to reflect the changes. We have also included patient treatment information in the main text to support the mutational signature analysis. We thank the reviewers for their comments that prompted us to make the changes.

4. Figure 2a is slightly hard to read – a suggestion would be to remove the box colors making a transparent box showing only the jitter points and shapes.

Figure updated as suggested.

5. Figure 2b. It is said in the results text that 7/8 high mutation samples have therapy-related mutational signatures, but it looks as though all samples have some non-zero value shading in cells for at least one therapy mut sig. If there is some exposure threshold that signatures must meet to be considered “present” in a patient, this needs to be made clear in the heatmap (either by zeroing-out sigs that don’t meet threshold, or an asterisk-like indication in cells that do meet threshold).

Done. We have added asterisks to Fig 2b to indicate signature weight greater than 0.25.

6. It is interesting that there are some PDX-specific oncogene or tumor suppressor gene mutations (Supplemental Figure 4d-e). Were they all predicted to be oncogenic/likely oncogenic or are they VUS (eg OncoKB)? Perhaps an updated figure which only includes oncogenic/likely oncogenic (eg predicted by OncoKB) would be more functionally relevant if these are mixed. Are these oncogene clonal mutations present at subclonal levels in the PT tumor even below variant calling criteria in the PT tumor (ie, did the authors do a deep dive outside of 585, even in the absence of additional deep sequencing)? Likewise, the authors can investigate this from the other direction, but the unexpected finding would be a novel oncogene mutation in the PDX which never existed in the PT tumor and what is the mechanism of this?

Using oncoKB-annotator, we identified 46 oncogenic or likely oncogenic mutations, 30 of which were found in PT-PDX pairs. Among the 30, 28 were shared between PT and PDX. The other two mutations were identified as likely oncogenic (*MUTYH*, 529_PDX; *MYCN*, 1823_PT). We have added these oncoKB annotation results in Supp Fig 4c,f (highlighted by asterisks).

The reviewer raised a longstanding question about the detection sensitivity of PT specific mutations in the matched PDX, or vice versa, PDX specific mutations in the matched PT. We did have a rescue strategy in our analysis: once we found a mutation unique to PT or PDX, we went back to raw output of each mutation caller and rescued the mutation if any caller identified it. This way, we in total rescued 37 mutations (2% of all mutations).

To further address the reviewers’ question, we did a supervised analysis for all the remaining PT or PDX specific mutations. Briefly, for any PT or PDX specific mutation, we searched for the mutation in the matched sample by piling up the nucleotides from sequencing reads at the mutation site. We found evidence (at least two reads carrying the mutant allele) for only 2 PDX specific mutations in the matched PT, and found no evidence of PT specific mutations in the matched PDX. This is a very small number considering that we have 294 PT or PDX specific mutations. These two mutations indeed have very low VAF in the PT (0.024 and 0.014) despite being clonal in the PDX. Encouragingly, their rediscovery lends further support to the evolutionary pattern for the corresponding PT/PDX pairs, both of which were classified as clone sweeping (529 and 1753. Clone sweeping: PDX derived from a PT subclone).

We have added these new data to the revised manuscript.

7. The “clone sweeping” examples vary from expansion of one subclone or many subclones, so the text can be that this is defined as at least one subclone expanding from PT to PDX.

Revised.

8. Figure 3g would likely be easier to interpret if it were shown as a box/dot/violin plot.

Done.

9. Lines 210-211: a correlation test should be performed to show statistical evidence for PT genetic heterogeneity being inversely associated with mutational similarity.

Done. Results are shown below ($\rho=0.43$, $p=0.034$, Spearman correlation). Figure R2 has been added to the manuscript (now Supplementary Fig. 7a).

Figure R2. The correlation between mutation similarity and PT heterogeneity. Each dot represents a paired sample. The correlation coefficient was calculated by Spearman’s correlation.

10. Lines 306-307: Should this say “hepatoblastoma and Wilms tumor samples showed significant intra-lineage correlations” (remove “Except for sarcoma”).

Done.

11. The observation that more mutationally diverse PT tumors yield PDX with differing genetics makes sense, but it does appear that 2/3 branch seeding models (1957 and 1823) were derived from PTs treated with chemotherapy, so this may not be unexpected. Additionally, can this be a tumor+chemotherapy-specific phenomenon? It appears that 2 hepatoblastoma models pre-therapy nicely recapitulated PT tumors, but the two with chemotherapy did not. In other cancer types, there is not a similar trend, albeit overall numbers are low. Were the retinoblastoma PTs treated with a different agent?

The reviewer made a great observation. We also noticed that the two chemo-treated hepatoblastomas showed the pattern of clonal selection, in contrast to the two without chemotherapy. We agree with the reviewer that chemotherapy could have played a role, given that chemotherapy as an external evolutionary pressure may increase the genetic heterogeneity of the PT. However, there is no enrichment of pre-chemotherapy between group 1 and group 2 tumors ($P=1$, chi-square test): 31% (5/16) of group 1 tumors versus 29% (2/7) of group 2 tumors received chemotherapy. Based on these results, we refrained from associating chemotherapy to the evolutionary patterns. We have added the chi-square test results to the revised manuscript.

Patient 1957 received AHEP0731 regimen T (cisplatin, fluorouracil, and vincristine); patient 585 received regimen AHEP1531 Group D (cisplatin, doxorubicin).

- Molecular Methods

12. The kits or protocols used for DNA and RNA extraction are not described in the Methods text.

Added in Method. Genomic DNA was extracted with DNeasy Blood & Tissue Kit (QIAGEN) and RNA was isolated using RNeasy Mini Kit (Qiagen).

- Computational Methods

13. “NGSCheckMate9 (v1.0.0) was applied to both DNA and RNA sequencing data to make sure matched samples were from the same patient.” This reads as if PDX DNA and RNA were checked for matching, but was this also done between PDX and patient to ensure PDX/patient matches?

We apologize for the confusion. What we did was exactly what the reviewers asked: we checked the matching between PTs and PDXs using both DNA and RNA sequencing data (Supplementary Fig 1). In the revision, we have reworded the sentence in Method to improve clarity.

14. The text states that “The BAM file of human reads was converted to FASTQ format using Samtools7 (v1.14) for further analyses”. This seems incorrect; was there another reason for re-generating fastq files with human-only reads?

To remove mouse reads from RNAseq data, we used an alignment-based tool called disambiguate. The input for disambiguate was the BAM file that contains mapped reads, because disambiguate differentiates human reads from mouse reads using mapping quality. The unmapped reads were not included in these BAMs. After removing mouse reads, the unmapped reads had to be recollected for fusion identification because they contain critical evidence, i.e., junction-spanning reads. We converted BAMs back to FASTQ so that we can merge them with the unmapped reads, and re-run fusion detection tools, both of which (STAR-fusion and PRADA) prefer FASTQ as input. We apologize for any confusion. In the revision, we have added these clarifications in Method.

15. “High confidence somatic mutations were identified as those that were called by at least two callers.” Were there any hotspot mutations missed by two callers, or were these checked, for example using the MSKCC hotspot database: <https://www.cancerhotspots.org/#/home>

We appreciate the reviewer’s concern. It is routine to use multi-callers for mutation detection. For instance, the TCGA MC3 effort used 7 callers (Ellrott et al. Cell Systems, 2018. PMID: 29596782). To address reviewer’s concern, we compared mutations downloaded from the MSKCC hotspot database with the mutations that have been filtered out. Of the 3554 somatic mutations that were called by only one caller, only one point mutation (*NUP93*, E14V) and two indels of *CTNNB1* were documented in the MSKCC hotspot database. As the reviewer knows, indels are notoriously difficult to identify from short-read sequencing data. Thus, the number of missed mutations is negligible. We note for the reviewer that in our analysis, we had taken additional steps to detect indels specifically for *CTNNB1* in hepatoblastoma samples because it is known that *CTNNB1* frequently harbors indels in this disease (Sumazin et al. Hepatology, 2017. PMID: 27775819). And we indeed detected one of these two hotspot indels in our hepatoblastoma samples. We have added these discussions in the revised Method.

16. What was the genetic sex makeup of the 40 PON samples? Is it balanced to ensure quality calls in XY chromosomes? What was the accuracy of the calls using 3/3 algorithms in tumor only samples? Did this result in a loss of any oncogenic calls?

The Panel of Normal included 17 female and 23 male samples. The male to female ratio (1.35:1) reflects the overall patient cohort (1.3:1).

We agree with the reviewer that the 3/3 algorithm is stringent. However, as the reviewer well understands, there are few established protocols for calling somatic mutations in model systems for which the matched normal is often unavailable. Loosening the stringency is much less desirable because it would inflate artifacts. In our study, if we use 2/3 approach for samples without germline, we could identify roughly twice as many mutations as the 3/3 approach. It is hard to know if these additional mutations are bona fide because there is no ground truth to compare with.

Several observations made us feel confident about our mutation data. First, the mutation rates of several cancer types are highly consistent between our dataset and published datasets. We refer the reviewer to **Table R2**. Second, known driver genes also show consistent mutation frequency with the literature. **Table R1** shows the mutation frequencies of known driver genes in our PDXs and in published datasets, including both PDX datasets (Murphy et al. and Rokita et al.) and patient tumor datasets (others). Although it is difficult to evaluate mutation frequency with small sample sizes, our mutation frequencies are generally consistent with those published in the literature. KIT in germ cell tumor is an exception. However, a close look suggests that KIT mutations were found in germinomas by the cited studies. Germinoma is not represented in our cohort.

	Gene	Grobner et al. Nature, 2018 (n=51)	PeCan2 (st.jude cloud, n=138)	Gadd et al. Nat Genet, 2017. (n=768)	Rokita et al. Cell Rep, 2019 (PDX, n=13)	Murphy et al. Nat Commun (PDX, n=45)	Our PDXs (n=13)
Wilms Tumor	CTNNB1	4%	6%	12%	15%	22%	15%
	SIX2	/	3%	3%	/	4%	10%
	NONO	/	3%	2%	/	4%	10%
	TP53	8%	19%	12%	23%	16%	/
	WT1	2%	3%	6%	8%	16%	/
	DROSHA	8%	7%	10%	15%	13%	/
	DGCR8	10%	3%	4%	/	7%	/
	SIX1	10%	3%	4%	/	7%	/
	Gene	Grobner et al. Nature, 2018 (n=42)	PeCan2 (st.jude cloud, n=151)		Rokita et al. Cell Rep, 2019 (PDX, n=36)		Our PDXs (n=11)
Osteosarcoma	RB1	5%	17%		17%		18%
	TP53	8%	20%		28%		9%
	ATRX	13%	19%		8%		9%
	ARID1A	5%	2%		/		9%
	SYNE2	/	4%		17%		18%
	Gene	Grobner et al. Nature, 2018 (n=16)	Hirsch et al. Cancer Discov, 2021 (n=65)	Sumazi et al. Hepatology, 2017 (n=88)			Our PDXs (n=11)
Hepatoblastoma	CTNNB1	19%	94%	89%			64%

	NFE2L2	/	6%	5%	9%
	DDX3X	/	3%	/	9%
	LEF1	/	3%	/	9%
Germ Cell Tumor	Gene	Kubota et al. Commun Bio, 2020 (n=51)	Shen et al. Cell Rep, 2018 (adult, n=137)		Our PDXs (n=9)
	KRAS	4%	13%		11%
	KIT	12%*	18%*		/

Table R1. Mutation frequency of known cancer drivers in 4 cancer types where we have relatively good sample sizes. Driver genes are defined by studies cited in the table.

*KIT mutations were found exclusively in germinomas in Kubota et al. Germ cell tumors in our cohort are of the testis and ovary.

We respectfully emphasize to the reviewer that 19 of 23 PT/PDX pairs had matched normal DNA; thus, the most important findings including the PT/PDX similarity and the evolutionary patterns are minimally impacted by the lack of germline samples.

We have included Table R1 in the revised manuscript (now Supplementary Table 3).

17. Mutational signature analysis: it is unclear what is meant by “We excluded ...signatures with the highest weight < 0.25 for further analysis.” Were mean signature weights recalculated using only the remaining signatures? (ie- if when using “all signatures ABCDEF”, Signature E had <0.25 in all samples, then was deconstructSigs rerun for only ABCDF signatures?) Additionally, it is common to filter out any signatures in patients with exposure weight < 0.06 (i.e., set them to zero). The results text and corresponding figure for this analysis make it unclear if this was performed.

We apologize for the confusion. For signature analysis, we first filtered out samples with low mutation counts. In the updated analysis, 58 samples with less than 20 mutations were excluded. We consider a minimum weight of 0.25 as evidence of robust detection of a signature, and removed samples where no signature had a weight higher than 0.25. Fourteen samples passed these filters in the final results (see Figure R1 and Figure 2b). No signature score scaling was done. In the revision, we have edited Method to improve clarity.

18. “The cancer gene list was downloaded from Cancer Gene Census17.” The authors can include this gene list as a supplemental table since this can change over time.

Done. The list is now included as Supplementary Table 5.

19. It is stated in the Methods that differential gene and pathway expression analyses were performed, but these results are not presented (gene) or shown in reduced form in Supp. Table 5 (enriched pathways). These results should either be presented in full in a supp. table AND discussed in the text, or the methods should not be included.

Done. We have removed these texts in the revised manuscript. In our initial pathway analysis, we found abundant immune-related pathways enriched with differentially expressed genes, but we chose not to show them because this was largely redundant with the GSEA results. We thank the reviewer’s scrupulous reading of our manuscript.

20. For differential expression analysis, the authors state that a multifactor analysis was performed but do not indicate if any additional covariates were included in the model (tumor type, treatment status, etc).

See our response to the above comments.

- Reproducibility

21. The code used to generate the results has not been shared within the manuscript text. The authors should share all computational analyses used in this manuscript via Github or some other repository in order to enable reproducible analyses of these data.

We did include the link to the Github repository in the software policy document in the initial submission (<https://github.com/fnhe/PediatricSolidTumorPDX>). We agree with the reviewer that this link should have been included in the main text (see data availability). We have hence included it in the revision.

Reviewer #2 (Remarks to the Author): expertise in solid tumour model development

In their manuscript titled “Genomic profiling of subcutaneous patient-derived xenografts reveals immune constraints on tumor evolution in childhood solid cancer,” He et. Al. describe their genomic profiling of a small cohort of pediatric solid tumors (90). Their engraftment rates were similar to several previous publications. Only 68 tumors were genomically profiled with low pass WGS, WES and RNA-seq. Only 27 of those 68 had the matched patient tumor and 40 of the 68 had the germline. With these limited data, they report clonal enrichment as has been shown in multiple studies previously. They claim that the clonal selection may be due to anti-tumor immunity in the patient relative to the PDX. The evidence for this comes from the observation that in some PDXs, the minor clone in the patient predominates and those cells are more proliferative. Overall, this is a very small cohort with incomplete characterization and validation and the conclusions regarding anti-tumor immunity are not supported by the data.

We respectfully but strongly disagree with the reviewer that (1) our cohort is very small, and that (2) our conclusion is not supported by the data. In the following rebuttal, we will demonstrate that our sample size is comparable to published high-profile studies, and our conclusion is supported by the data (“based on multiple lines of evidence” per Reviewer 1).

Specific Comments:

1. A major limitation is the lack of patient tumor and germline for all of their cohort. The patient tumor and germline should be used with WGS, WES and RNA-seq to identify somatic lesions. The PDX and patient tumor should be used to assess clonal heterogeneity. Without all 3 samples (patient, germline, PDX) the impact of the study is minimal. Based on Fig. 1, there are only 22 with all 3 samples. Among those 22, 8 are Wilms, 2 are clear cell sarcoma, 1 is neuroblastoma, 3 are osteosarcoma, 4 are hepatoblastoma and 3 are germ cell tumor. It is very difficult to make any conclusion with such small numbers of tumor samples.

Childhood cancers make up about 1% of all cancer diagnoses in the US. Among them, about 40% are blood cancers, leaving the total incidence of solid childhood cancers to be around 0.6% of all cancer diagnoses. Please note that this 0.6% comprises many different entities; the central nervous system alone has more than 100 known cancer types (Capper et al. Nature, 2018. PMID: 29539639). Essentially, solid cancers in children are rare cancers. This rarity imposes an enormous challenge for basic, preclinical, and clinical research. That is why collaborative groups such as COG (Children’s Oncology Group) have been formed to coordinate clinical trials because it is difficult, if not impossible, for single centers to recruit enough cases for trials. PDXs are

important for the same reason because they can be used to prioritize trials and preserve tumor tissue.

Below is a table comparing sample sizes between our dataset and previously published studies. We are not aware of other larger PDX datasets for childhood solid cancer. Given that the overall engraftment rate is ~48%, establishing 68 PDXs would take around 140 patient tumors. This is an enormous size for pediatric solid cancer. Of note, the Rokita et al. study is a consortium effort on sequencing PDXs used in NCI's Pediatric Preclinical Testing Consortium (PPTC, since 2004). Even with the consortium effort, the sample size appears small compared to many adult cancer studies. This again shows the challenges with childhood cancer research and the resource value of our dataset.

Study	Molecularly profiled Solid tumor PDX	PDX/PT pair	Germline	Cancer type
Stewart et al. Nature 2017 PMID: 28854174	67	51*	51	Diverse
Rokita et al. Cell Rep, 2019 (PPTC) PMID: 31693904	171	0	0	Diverse
Our cohort	68	27	40	Diverse

Table R2. Comparison of Cohort size of our dataset and other two published datasets. * Clonal analysis was done on 42 of the 51 PT/PDX pairs.

The lack of PT or germline samples is not without reasons. Because these tumors are so rare, tumors are typically cut in several pieces for various purposes including pathology review, banking, and research. Samples we receive are usually very small in size, some from biopsies. To ensure successful generation of PDXs, we implant a tumor sample into 2-5 mice. This typically exhausts the tumor tissue. We send PTs to sequencing only when there is tissue left. This problem is not unique to us. In a landmark paper (Stewart et al. Nature, 2017. PMID: 28854174), the authors noted in the supplementary discussion that “The biggest challenge of producing O-PDX tumors was procuring enough tissue. For biopsy specimens or small tumor samples, there is rarely enough tissue for injection.”

In summary, the relatively small sizes of childhood cancer cohorts compared with adult cancers stem from their rarity, a reality that highlights the value and impact of this resource. Besides its resource value, we show in our paper how this resource can be used to shed light on fundamental questions about PDXs, i.e., what causes the genomic disparity between PTs and PDXs. We respectfully remind the reviewer that our clonal analysis is supported by multiple lines of evidence derived from orthogonal data. These insights are novel, and the implications are much beyond childhood cancer PDXs.

In the revised manuscript, we have added texts to emphasize the rarity of childhood solid cancers.

2. It is not clear why low pass WGS was performed rather than deeper WGS and there was no discussion of validation using custom capture and deep sequencing or at least cross validation of the WES and RNA-seq.

We used low pass WGS and WES because the two methods give us complementary, high-quality data. Low pass WGS provides genome wide coverage thus is more suitable for copy number

profiling. For mutation calling, WES provides a depth of 300-400x. This depth allows for detection of somatic mutations at a very low cellular frequency. Deeper WGS can do both copy number profiling and mutation calling. But at the depth of 300-400x the cost is unsustainable for us. In our view, both WES and low pass WGS are mature, reliable sequencing methods, and there is no reason we cannot use them.

Regarding the reviewer's comment on mutations validation, we did have deep sequencing data for 585_PT, as the reviewer noted below. More importantly, Illumina WES is technologically very mature. We respectfully point out to the reviewer that the consistency between our dataset and the published datasets (Ma et al. Nature, 2018; Grobner et al. Nature, 2018) in mutation rates (see Table R2), and the expected mutation frequencies of driver genes in selected cancer types (see Table R1) provide strong validation for our mutation data.

To address the reviewer's concern on absence of validation, we cross-compared mutations from WES with RNAseq and low pass WGS data. Of the 1,787 mutations called from WES, 211 have coverage ≥ 10 in low pass WGS (minimum mapping quality 20), 184 of which show at least one read covering the mutant allele (validation rate 87%). The unvalidated mutations (n=27) show an average VAF of 0.15 (median 0.13; mean 0.15). At this VAF, we would only expect 1-2 reads at the sequencing depth of 10. In RNAseq, 219 mutations show coverage ≥ 10 , 194 of which show at least one read covering the mutant allele (validation rate 89%). Similarly, the unvalidated mutations have low VAFs (median, 0.11; mean, 0.15). If we combine RNAseq and low pass WGS, 388 mutation sites have coverage ≥ 10 in either data type, and 356 of them show at least one read covering the mutant allele (validation rate 92%). The average VAF of validated mutations is 0.34, compared with 0.13 of the unvalidated mutations.

In addition, we now add deep sequencing data on 585_PDX (average 7000x). Of the 55 point mutations from WES in this sample, 54 were captured in deep sequencing, and all 54 were validated. A similar result was described in the manuscript for 585_PT (6/6 mutations validated by deep sequencing). Taken together, these results show our WES-based mutations are highly confident.

We have added descriptions on the new validation data in the revised manuscript. The deep sequencing data has been provided in Supplementary Table 2.

3. The mutation rate and mutation signature analysis were limited by sample size(???) and there are much larger studies that have been published on patient tumors and PDXs with rigorous validation and deep sequencing.

We would appreciate the reviewer's reference to the specific childhood cancer PDX studies that have much larger sample sizes. We are not aware of such studies except those listed in Table R2.

We apologize for not understanding how sample size can limit mutation rate analysis, which is calculated on a per-sample basis. We respectfully point out that our mutation rates are consistent with results from the recently published pan-childhood cancer studies (Grobner et al. Nature, 2018, PMID: 29489754. Ma et al. Nature, 2018, PMID: 29489755). The following table compares the median number of exonic mutations of three cancer types shared between our dataset and the two studies. As the reviewer can see, the mutation rates are highly similar. These results lend strong support to our mutation analysis.

	Our data	Grobner et al.	Ma et al. (WGS)	Ma et al. (WES)
Wilms tumor	7	11	7	7
Neuroblastoma	15	17	19	16
Osteosarcoma	22	23	31	21

Table R3. Comparison of median mutation rates between our study and two recently published studies across three cancer types that were shared among the three studies. For WGS and WES mutation data from *Ma et al.*, we inferred the mutation number based on 39Mb exonic region (same as our data) and the mutation rate listed in their manuscript.

Mutation signature analysis is limited by mutation counts per sample, not by sample size. Our mutation signature results are highly consistent with patient treatment history when the tumor has a decent mutation count: of the seven samples that carried a platinum-based drug signature (SBS31 and SBS35), all received the platinum-based drug cisplatin. This consistency strongly supports our mutation signature analysis. We refer the reviewer to Figure R1.

4. It is interesting that they did perform capture enrichment and deep sequencing on a single patient tumor that had low PDX similarity. However, they did not perform the same analysis on the PDX. The proper way to do the study is to design capture probes for all somatic SNVs across the genome in each patient-PDX pair and then do deep sequencing on all SNVs on both samples. Using this approach, they can identify rare clones in both the patient tumor and the PDX.

We appreciate the reviewer raising this issue. In the initial submission, we performed deep sequencing on 585_PT but not 585_PDX, because in our view deep sequencing of the PDX sample provides little biological insight--it is pointless to know if a PT specific mutation exists in the PDX at an ultralow frequency. Nonetheless, for the revision, we have done deep sequencing on 585_PDX (average depth, 7000x). As we mentioned in response to comment #2, the deep sequencing data validated all 54 mutations that were captured in the assay. None of the PT specific mutations were found in the PDX deep sequencing data and vice versa.

We wish to discuss with the reviewer about the use and interpretation of deep sequencing data.

First, “deep” sequencing must be considered a quantitative but not qualitative measurement. There is no golden-standard depth of coverage for deep sequencing data. For example, compared to 50x, both 400x and 4000x can be considered “deep”. The difference is the detection sensitivity. This is very important, because at any of these depths, the absence of a mutation in deep sequencing data does not prove the absence of the mutation in the sample; it only proves the absence of the mutation at the specified sensitivity. The average depth of our WES data is 317x (min, 182x; max, 795x). At this depth, we can detect variant alleles at a fraction as low as 1.6% (assuming minimum 4 mutant reads and 80% purity). This is equivalent to a cellular frequency of 3% (assuming neutral copy number, heterogeneous mutation). We think this is an acceptable sensitivity.

Second, the scope of deep sequencing data is bounded by WES/WGS. Deep sequencing data does not identify new mutations. Rather, their purpose is to validate mutations identified by WGS or WES. In the context of PT vs. PDX, if a mutation is not detected in WES or WGS, deep sequencing cannot be used to call it for a simple reason: at 1000x, given the Illumina sequencing error rate (0.1%-0.5%, Stoler et al. PMID: 33817639), we would expect 1-5 variant reads at any nucleotide site. Moreover, deep sequencing is typically capture enriched, meaning by design only mutation sites identified by WGS/WES are effectively captured. Thus, if the WES or WGS does

not identify mutations that are indicative of a rare clone, deep sequencing certainly cannot identify them.

Third, the sample 'PT', as we call it, is not the tissue origin of the PDX. The tissue origin of PDX has been exhausted to generate the PDX. As such, its genetic makeup is impossible to know. The 'PT' sample is taken from a different section of the patient tumor. How PT genetically relates to the tissue origin of the PDX is determined by intratumoral heterogeneity (see **Figure R4**). If deep sequencing of the 'PT' detects PDX specific mutations, most likely these mutations also exist in the PDX's tissue of origin. However, absence of such mutations in deep sequencing data does not suggest that these mutations are absent in the PDX's tissue origin, and hence, the patient tumor. For this reason, deep sequencing does not provide definitive evidence for the presence or absence of PDX specific mutations in the patient tumor. This is also why we think treatment related mutational signatures are so important. Because mice were not treated with chemo-agents, the observation of chemo-related signatures in PDXs can be only explained by the inheritance of mutations from the tissue origin, no matter whether these mutations are seen in the 'PT.'

Here we are not denying the value of deep sequencing. It provides higher sensitivity for detecting PDX specific mutations in PTs and can potentially prove these mutations pre-exist in the patient tumor (Stewart et al. Nature, 2017. PMID: 28854174). However, it has also been shown that a substantial proportion of PDX specific mutations are not found in the PT by deep sequencing (Murphy et al. Nat Commun, 2019. PMID: 31862972). As we pointed out in the discussion above, interpreting this negative result carries certain ambiguity. Thus, deep sequencing has its limitations and is not used in every PT/PDX study (examples include a study by NCI PDXNet Consortium. Sun et al. Nat Commun. 2021. PMID: 34429404).

In the revision, we have added more discussion on deep sequencing in Methods.

Figure R4. Schematic of PT, PDX, and PDX tissue origin (PT-TO). PT denotes the patient tumor sample that is genomically profiled. However, PT is distinct from the tissue origin of the PDX, which is denoted as PT-TO. PDX here denotes early passage PDX tumors such as used in our study.

5. A germline sample is required to serve as a reference to identify somatic mutations. It is not clear how they could conclude that mutations were clonal without doing the target enrichment and capture for all the SNVs across all the tumors. Subclonal SNVs may be lost in the shallow WGS and the exonic SNVs may have functional significance so interpreting clonal architecture in WES alone must take into account the rare SNVs and the possibility that they provide a selective growth advantage or disadvantage.

We apologize for not understanding the reviewer's comments. We used WES and never used shallow WGS for mutation calling, thus we are confused by the reviewer's comment "Subclonal SNVs may be lost in the shallow WGS."

We are also baffled by the comment "It is not clear how they could conclude that mutations were clonal without doing the target enrichment and capture for all the SNVs across all the tumors." We speculate by "all the tumors" the reviewer meant paired PT and PDX samples. Target enrichment and sequencing of paired PT and PDX does not inform the clonality of a mutation; the presence of a mutation in both PT and PDX does not suggest the mutation is clonal. A subclonal mutation in the PT can be clonal or subclonal in the PDX, despite being present in both samples.

We respectfully bring the reviewer's attention to the fact that computational methods for inferring mutation clonality from DNA sequencing data have been established for more than 10 years. Examples include ABSOLUTE, published in 2012 (Carter et al. Nat Biotech, 2012. PMID: 22544022); Pyclone, first introduced in 2012 and formally published in 2014 (Shah et al. Nature, 2012. PMID: 22495314. Roth et al. Nat Methods, 2014. PMID: 24633410). These methods infer mutation clonality based on mutation variant allele fraction while controlling for copy number alteration and tumor purity. They do not need sequencing data from multiple samples. In our data, we found 88% of PT clonal mutations were observed in the PDX, while only 22% of PT subclonal mutations were observed in the PDX (first paragraph in the 'evolutionary pattern' section). We explicitly reported these numbers in the manuscript because the contrast (88% vs. 22%) is in accord with the expectation that clonal mutations are more likely to pass on than subclonal mutations, thus providing support for our inference of mutation clonality.

We agree with the reviewer that having a germline sample is important for identifying somatic mutations but disagree that a germline sample is absolutely required for mutation calling. There are scenarios where it is simply impossible to get germline samples, particularly for model systems. For instance, no germline sample is available for established cancer cell lines included in Cancer Cell Line Encyclopedia (CCLE), and yet CCLE, including its mutation data, is a vital resource for the cancer research community. None of the PDX models profiled by PPTC (Rokita et al. Cell Rep, 2019) has the matched germline. But these models have been yielding critical preclinical insights that guide clinical trials on childhood cancer. We respectfully point out to the reviewer that only 4 of the 23 PT/PDX pairs do not have germline samples, and none of the 4 pairs showed the clonal selection pattern. Thus, the lack of germline samples should have a very small impact on our core messages including the PT/PDX similarity and the evolutionary patterns.

In the revision, we have annotated PT/PDX pairs in each figure whenever they do not have matched germline. This annotation is to remind readers of this caveat. The affected figures are Fig. 2c, 4a.

6. The clonal selection has been reported previously but there is no biologic or mechanistic data presented here to indicate that it is related to antitumor immunity rather than a stochastic event or some other mechanism. Indeed, if a minor clone overtakes the PDX, it is predicted to be more proliferative than in the patient but this does not prove anti-tumor immunity in the patient.

We are glad the reviewer agrees with us on the one key piece of data leading to our main conclusion. We kindly point the reviewer to Fig 4c,d,e for the evidence leading to our conclusion of anti-tumor immunity in patients. Briefly, we show in Fig 4c,d,e that patient tumors that undergo clonal selection have stronger expression of gene signatures related to innate immunity and antigen presenting cells, and their corresponding PDXs express more clonal neoantigens.

Reviewer #3 (Remarks to the Author): expertise in clone sweeping and genomics analysis

The authors generated patient-derived xenografts (PDXs) data from 65 pediatric tumors with various cancer types. Mutation similarity between the patient tumor (PT) and PDX was analyzed. The number of PDX datasets that were generated in this study is very large, and it will be a great resource. Also, the pattern of evolution in PDX is still largely unknown, so the findings are potentially useful. However, the results need to be more carefully interpreted.

We thank the reviewer for acknowledging the value of this resource and the novelty of our study.

1. In this study, only a single section of PT was sequenced, and similarities of mutation, copy number, and transcriptomics profiles between PT and PDX were analyzed. However, the observed difference can be due to sampling error, i.e., subclone/clone used for the xenograft was not sequenced in the PT data. Sequencing more sections from the PT may change the amount of observed similarity between PT and PDX. Ideally, sequencing more than one sample is necessary for this type of data analysis. Actually, sequencing multi-sections is very common in the analysis of tumor evolution, as sampling errors due to sequencing a single sample are a well-known problem.

We agree with the reviewer that sampling can affect PT/PDX similarity. In the initial manuscript, we had briefly discussed the potential impact of sampling. Samples we receive are usually very small in size, some from biopsies. To preserve tumor tissue, we implant tumor tissue into 2-5 mice. We rarely have an additional tumor block for sequencing. This problem is not unique to us. In a seminal paper by Stewart et al. (Nature, 2017. PMID: 28854174), the authors noted that “The biggest challenge of producing O-PDX tumors was procuring enough tissue. For biopsy specimens or small tumor samples, there is rarely enough tissue for injection.” (O-PDX stands for orthotopic PDX).

Figure R5. Schematic of additional PDXs. For the revision, we identified additional PDXs that were directly established from the patient tumor. The 'PT' sample and the cellular origin of these PDXs each represent one sampling of the patient tumor.

We took an alternative approach to address the reviewer's comment. As we mentioned above, we implant tumor tissue into multiple mice. In some cases, we get multiple PDXs, each derived from a different section of the primary tumor (please also see **Figure R5**). In total, we identified 7 more P1 PDXs (the tumor grown in mice after initial implantation) that were established from the same patient tumors. One PDX had to be excluded due to high mouse cell contamination, leaving the total to 6. Additionally, we identified a second tumor block for 1795_PDX (noted as 1795_PDX-2). Comparing these additional PDXs to the matched PT, we indeed see variation in mutational

similarity as the reviewer suggested (**Table R4**), but the similarities are generally consistent. Copy number patterns between PDXs and the PT are highly correlated (**Figure R6**), corroborating our conclusion that copy number events are mostly early and are preserved in PDXs (see also Woo et al. Nat Genet, 2021. PMID: 33414553). We have added these new data to the manuscript (Supplementary Fig. 4g; Supplementary Fig. 9d).

	Shared	PT	PDX	Similarity
1795_PDX	5	7	9	0.45
1795-A_PDX	5	7	7	0.56
1795_PDX-2	5	7	7	0.56
1826_PDX	18	21	21	0.75
1826-A_PDX	20	21	26	0.74
1913_PDX	8	19	15	0.31
1913-A_PDX	8	19	23	0.24
1913-B_PDX	9	19	20	0.30
1932_PDX	4	6	5	0.57
1932-A_PDX	4	6	8	0.40
1932-B_PDX	4	6	7	0.44

Table R4. PT and PDX mutational similarity using different PDXs established from the same patient tumors. The first PDX in each group is the PDX sample analyzed in the initial dataset. A and B are additional P1 PDXs. 1795_PDX-2 is a second block of 1795_PDX. ‘Shared’ indicates the number of shared mutations between matched PT and PDX. ‘PT’ and ‘PDX’ columns indicate the total number of mutations detected in the PT and PDX, respectively.

Figure R6. Copy number correlation between PT and PDXs. The correlation was calculated the same way as in Fig 5 in the manuscript.

In practice, which section to take from patient tumor for the purpose of either engraftment or PT profiling is usually quite random. This randomness, or sampling error as the reviewer put it, is universal in genomic profiling of bulk tumors. The way to obtain reliable patterns over this randomness is to increase sample size. Multi-sector sequencing suggested by the reviewer increases the sample size on the single tumor level. While this can be very insightful for understanding a particular tumor, extending the insights to other tumors would require multi-sector sequencing of those tumors. This is challenging even impractical for pediatric cancer, especially if many tumors are needed for sufficient statistical power. Alternatively, one can look for patterns through a cohort, because even events of low probabilities will manifest if the cohort has a proper

sample size. Since these patterns transcend the identity of each sample, we would also expect to observe such patterns across multiple cohorts.

To demonstrate this, we show PT/PDX similarity from our study (**Figure R7a**), the study by Stewart et al. (Nature, 2017. PMID: 28854174) (**Figure R7b**), and the study by Murphy et al. (Nat Commun, 2019. PMID: 31862972) (**Figure R7c**). The Stewart et al. study examined 67 orthotopic childhood cancer PDXs, and the Murphy et al. study examined 45 subcutaneous childhood Wilms tumors. As the reviewer can see, the patterns are generally similar, with some pairs showing very low mutational similarity. We think this data support the idea that though every PT/PDX pair is subject to sampling randomness, when a large number of pairs are profiled, they can inform on the general patterns such as PT/PDX genetic similarity and evolutionary patterns. We indeed made recurrent observations with Stewart et al. For instance, we found high PT/PDX similarity in osteosarcoma (see Fig 4a), so did they; we found longer engraftment time associated with clonal selection (see Fig 3d), and they noted longer engraftment time in tumors (neuroblastoma) that show worst clonal preservation (please see our response to comment #5).

Figure R7. PT/PDX mutational similarity. (a) from our cohort. (b) from Stewart et al. (c) from Murphy et al. Mutation data for the two other studies were downloaded from their published Supplementary tables.

2. Three evolutionary patterns were identified, i.e., clone retention, clone sweeping, and branch seeding. However, this classification can be also affected by sampling errors of clones in PT.

Please see our response above. We also kindly remind the reviewer that our evolutionary patterns are corroborated by their correlation with orthogonal data, including tumor telomere lengths (Fig. 3g), engraftment time (Fig. 3d), and expression profiles (Fig. 4b-e).

3. The low mutational similarity was concluded to be correlated with the high genetic heterogeneity of the PT without reporting clones within PT. Clones and clone phylogenies should

be inferred, which is a common practice. Please note that clone prediction from a single tumor sample is not reliable, so caution is necessary.

We defined genetic heterogeneity of a tumor as the fraction of subclonal mutations over all mutations. Thus, more subclonal mutations would indicate higher genetic heterogeneity. We did not map out clones or clone phylogeny because of low mutation counts in childhood cancer (in our cohort, median, n=14; mean, n=23). Though in some cases the clonal structure is clear (see Fig 3a-c for examples), the low mutation counts make it often infeasible to distinguish distinct clones with high confidence. Thus, we used this more robust way to quantify a tumor's genetic heterogeneity.

We agree with the reviewer that with single biopsy, it is difficult to distinguish between early mutations and dominant mutations, the latter of which could be specific to one specimen. To test the reliability of our mutation clonality prediction, we again used the additional PDXs that matched the four PT samples (**Table R4**). Our rationale is, if a PT clonal mutation passes on to one PDX, the mutation should also pass on to other PDXs due to its "clonal" nature. For the reviewer's convenience, **Figure R8** below shows the mutation dynamic between the PT and the original PDX. With the additional PDXs, we observed the following:

- In case 1795, 4 PT clonal mutations were observed in the original PDX (1795_PDX). All these 4 PT clonal mutations were also found in the additional PDX (1795-A_PDX).
- In case 1795, all 7 clonal mutations of the original PDX block (1795_PDX) were also observed in the second tumor block of the PDX (1795_PDX-2). The subclonal mutation was not observed in the second block.
- In case 1826, 17 PT clonal mutations were observed in the original PDX (1826_PDX), and all 17 were also observed in the additional PDX (1826-A_PDX).
- In case 1913, 6 PT clonal mutations were observed in the original PDX (1913_PDX), and all 6 were also observed in the two additional PDXs (1913-A_PDX and 1913-B_PDX).
- In case 1932, 3 PT clonal mutations were observed in the original PDX (1932_PDX), and all 3 were also observed in the two additional PDXs (1932-A_PDX and 1932-B_PDX).
- 17 subclonal mutations were identified across the 4 PTs. Only 2 of the 17 were observed in the additional PDXs, in stark contrast to the PT clonal mutations.

Figure R8. Mutation dynamic for the 4 PT/PDX pairs where more than one P1 PDXs were characterized. The figures were taken from Supplementary Fig. 5c. PDX here refers to the PDX used in the initial analysis.

Despite a small sample size, these results strongly support our prediction of mutation clonality. We gently remind the reviewer that across all PT/PDX pairs, we observed 88% of PT clonal

mutations in the PDX, compared to only 22% of PT subclonal mutations (see first paragraph in the ‘evolutionary pattern’ section in the initial submission). This is consistent with the expectation that clonal mutations in the PT more likely pass to the PDX.

We have made other intriguing observations from this new data. Both 1913 and 1932 were classified as clone sweeping, i.e., PDX main clone was derived from a PT subclone. The evidence for this classification is that in both cases, a PT subclonal mutation became a clonal mutation in the PDX (*LRP2* for 1913, and *BMP4* for 1932). Interestingly, the same *LRP2* mutation was identified in the two additional 1913 PDXs, and the mutation appears to be clonal (VAF 0.44 and 0.45 in PDXs vs. 0.09 in PT). Similarly for 1932, the *BMP4* mutation was observed in the two additional PDXs, also with much higher VAF (0.37 and 0.38 in PDXs vs. 0.12 in PT). Thus, the clone expansion (so is the evolutionary pattern) seems to be very consistent in these cases. This conclusion also holds for 1795 and 1826, both of which were classified as clone retention. The additional PDXs for the two cases well preserve the clonal PT mutations, supporting the evolutionary pattern that was defined based on the initial PDX sample. These observations address Reviewer’s comment #2.

Moreover, 7 PDX specific clonal mutations were also found in the two additional PDXs in 1913. This observation provides definitive evidence that these mutations preexist in the primary patient tumor, because the statistical odds for PDXs grown in different mice to acquire 7 identical mutations would be extremely small (virtually impossible given the size of the genome).

We have added these new results and data to the revised manuscript (Supplementary Fig. 4g). We thank the reviewer for the excellent question that prompted us to do these analyses.

4. Observed similarities of mutation, copy number, and transcriptomics profiles between PT and PDX are often not properly interpreted. For example, chemotherapy-related signatures were expected to be observed for PDXs when their corresponding PTs have those signatures, because mutations that occurred in PTs are inherited in their PDXs. To test if mice without treatment accumulate mutations under the chemotherapy-related mutational processes, mutations that are unique only in PDXs should be compared with those from PTs. But, only a few datasets have a sufficient number of mutations that are unique to PDXs, so I think mutational processes in PDXs cannot be elucidated from these datasets.

We thank the reviewer for understanding the challenge with rare cancer research. We appreciate the reviewer’s arguments about mutational signatures. In our manuscript, we were very cautious with interpreting the mutation signatures. In the fourth paragraph of the section “mutational similarity between PT and PDX,” we stated that “*The consistency in demonstrating chemotherapy signatures was **not necessarily driven by shared mutations** between PTs and PDXs. For 585 and 1957, PTs and PDXs had little overlap in somatic mutations (Fig. 2c). Thus, these data suggest **the related mutations in these PDXs** were inherited from the seeding PTs.*” By “*not necessarily driven*,” we acknowledge in some PT/PDX pairs, the signature consistency is driven by shared mutations, but meanwhile we point out that some cases are exceptions such as 585 and 1957. By “*...related mutations in these PDXs were...*” we confine our conclusion to the aforementioned PDXs, namely 585 and 1957.

We kindly note for the reviewer that for 585 and 1957, PTs and PDXs have very low mutational similarity: only 3% overlap for both pairs. However, 89% of mutations in 585_PDX and 77% of mutations in 1957_PDX are attributable to chemo-signatures SBS31 and SBS35 (Fig. 2b). Thus, it is impossible that the shared mutations drive the signatures.

To further demonstrate this, we repeated the mutational signature analysis on four samples with ≥ 10 PDX specific mutations (**Figure R9**). Fig R9a shows mutation signatures using all PDX mutations, and R9b shows mutation signatures using only PDX specific mutations. It is clear that both sets of mutations strongly indicate the presence of chemo-signatures in 1959_PDX and 585_PDX. The other two PDXs (529 and 1979) do not show conspicuous mutation signatures (no signature has weight > 0.25).

In the revision, we soften our statement and removed the sentence “The observation of chemotherapy-related signatures in PDXs is significant because the tumor-bearing mice were never treated with chemotherapy.” We also revised the sentence in Discussion related to mutational signatures to inform readers about the reproducibility of the chemo-signatures using PDX unique mutations. We have added Fig R9 to the revised manuscript (Supplementary Fig 9d).

Figure R9. Mutation signatures using all PDX mutations (a), or PDX specific mutations (b). We limited our analysis to the four PDXs that have at least 10 PDX specific mutations.

5. The evolutionary patterns were concluded to correlate with PDX engraftment time. However, the number of unique mutations within PDX is expected to increase as the engraftment time. The mutation rate should be computed and compared.

The reviewer raised an excellent point. To test this possibility, we correlated PDX engraftment time with PDX unique mutations. We didn't observe a significant correlation ($\rho=0.3$, $p=0.18$. **Figure R10a**). We then repeated the analysis by controlling for the number of mutations in PT (assume they are the baseline for each PDX); again, no significant correlation was observed ($\rho=0.34$, $p=0.13$. **Figure R10b**).

We noticed in the paper by Stewart et al. (Nature, 2017. PMID: 28854174), the authors reported that "Osteosarcoma had the best clonal preservation in O-PDX models and neuroblastoma had the worst. Neuroblastomas also had the longest engraftment time ..." This observation corroborates our finding that clonal selection is associated with prolonged engraftment time (Fig 3b), even though their observation was made on orthotopic grafting. Interestingly, we also observed excellent mutational similarity for osteosarcomas in our dataset (Fig 4a).

We have added **Fig. R10** in the revised manuscript (Supplementary Fig. 5e,f). We thank the reviewer for raising this great question.

Figure R10. The correlation between engraftment time (week) and the PDX private mutations in PT-PDX paired samples (a) and Wilms Tumor (b). Each dot represents one PDX sample. The correlation coefficient was calculated by Spearman's correlation.

6. Line178: The loss of PT clonal mutations can be also expected through copy number alteration (deletion). To reject this possibility copy number analysis is necessary.

Loss of PT clonal mutations is a feature of the 'branch seeding' evolutionary pattern. There are 3 PT/PDX pairs demonstrating this pattern, and we in total identified 13 clonal mutations unique to the 3 PTs. On the absolute copy number level, no clonal mutation was in deleted regions in the PDX. We have clarified this in the revised manuscript.

7. Line 213: How were major clones identified?

The 'major clones' in this context (paragraph starting at line 213 in initial submission) are to introduce the biological concept for our next analysis (Fig. 4b). It is a wording choice, not a technical term. We apologize for the confusion. However, the distribution of mutant allele fraction (Fig. 3b-c) shows clear clonal structure in group 2 samples.

REVIEWERS' COMMENTS

Reviewer #1 (Remarks to the Author):

The authors did a nice job responding to reviews and as a result, have improved the rigor of the manuscript. Before publication, the sequencing data should be deposited into dbGAP or other repository and the accession number listed in the manuscript. Otherwise, we have only minor comments below.

Figure 2a – line 133 reports a Wilcoxon rank sum test p-value of treated vs. treatment-naïve mutation rates, but this box plot does not report any associated test statistics or p-values. Also, were differences within tumor type tested for? It seems more appropriate to run analyses similar to what is shown in Figure 4A and 4B, where samples are compared within tumor type.

Figure 2b – The tumor types not present with the updated mutational signatures analysis should be removed from the legend for clarity.

Reviewer #2 (Remarks to the Author):

A quick review of the literature found resources (published and online) for over 500 PDXs have been published or made available representing more than 2 dozen different tumor types. The models and their associated data are all freely available to the community and many of the models have more in depth characterization. Indeed, all the associated data are freely available so the authors could have easily incorporated the larger cohort to boost their claims.

Reviewer #3 (Remarks to the Author):

I think the sequencing depths and methods (300x for WES and 4x for WGS) are the standard. In addition, targeted deep sequencing (5000-7000x) was performed to validate detected mutations, which is also a common practice. I do not see any issues with the sequencing. All concerns were addressed including Reviewer #2, and the manuscript was improved. I do not have further concerns.

Reviewer #1 (Remarks to the Author)

1. The authors did a nice job responding to reviews and as a result, have improved the rigor of the manuscript. Before publication, the sequencing data should be deposited into dbGAP or other repository and the accession number listed in the manuscript. Otherwise, we have only minor comments below.

The sequencing data have been deposited to European Genome-Phenome Archive (EGA). Accession number EGAS00001006710). This information is available in "Data Availability."

2. Figure 2a – line 133 reports a Wilcoxon rank sum test p-value of treated vs. treatment-naïve mutation rates, but this box plot does not report any associated test statistics or p-values. Also, were differences within tumor type tested for? It seems more appropriate to run analyses similar to what is shown in Figure 4A and 4B, where samples are compared within tumor type.

We have added cancer type specific comparisons to Figure 2a. Figure legend has also been updated to include the details.

3. Figure 2b – The tumor types not present with the updated mutational signatures analysis should be removed from the legend for clarity.

Done.

Reviewer #2 (Remarks to the Author)

A quick review of the literature found resources (published and online) for over 500 PDXs have been published or made available representing more than 2 dozen different tumor types. The models and their associated data are all freely available to the community and many of the models have more in depth characterization. Indeed, all the associated data are freely available so the authors could have easily incorporated the larger cohort to boost their claims.

The two links direct to the same data portal (CSTN, Childhood Solid Tumor Network data portal). This portal has been published in Stewart et al. Nature, 2017 (citation under "Resources" tab on the site home page). In our paper and previous rebuttal, we have repeatedly compared our results with those from Stewart et al., including engraftment rate, PT/PDX similarity, etc. (ref 21). Some important observations from Stewart et al., such as better clonal preservation in osteosarcoma, longer engraftment time in models with low clonal preservation, are also consistent with our data.

An important distinction between CSTN PDXs and our PDXs is that ours are subcutaneous PDXs, whereas CSTN models are orthotopic models, as we emphasized in the title and "Introduction" of the manuscript. This distinction should not be trivialized.

CSTN currently has 281 orthotopic PDX models (far less than 500), a significant increase from the original cohort reported in Stewart et al. (n=67). While most molecular data are indeed available for browsing (but not batch downloading), data for the newly added >210 models have never been peer reviewed. Without publication and peer review, it is unclear if sequencing data for these newly added models have been subject to the same processing and quality control criteria. Based on these reasons, we refrain from comparing our data with the data at CSTN.

Reviewer #3 (Remarks to the Author)

I think the sequencing depths and methods (300x for WES and 4x for WGS) are the standard. In addition, targeted deep sequencing (5000-7000x) was performed to validate detected mutations, which is also a common practice. I do not see any issues with the sequencing.

We thank this reviewer for endorsing us.